# Genetic diversity in terrestrial subsurface ecosystems impacted by geological degassing

Till L. V. Bornemann[1], Panagiotis S. Adam [1], Victoria Turzynski[1], Ulrich Schreiber[2],
Perla Abigail Figueroa-Gonzalez[1], Janina Rahlff [1,7], Daniel Köster [3], Torsten C. Schmidt[3,4], Ralf Schunk[5],
Bernhard Krauthausen[6] & Alexander J. Probst [1,4✉]

Earth's mantle releases 38.7 ± 2.9 Tg/yr $CO_2$ along with other reduced and oxidized gases to the atmosphere shaping microbial metabolism at volcanic sites across the globe, yet little is known about its impact on microbial life under non-thermal conditions. Here, we perform comparative metagenomics coupled to geochemical measurements of deep subsurface fluids from a cold-water geyser driven by mantle degassing. Key organisms belonging to uncultivated *Candidatus* Altiarchaeum show a global biogeographic pattern and site-specific adaptations shaped by gene loss and inter-kingdom horizontal gene transfer. Comparison of the geyser community to 16 other publicly available deep subsurface sites demonstrate a conservation of chemolithoautotrophic metabolism across sites. In silico replication measures suggest a linear relationship of bacterial replication with ecosystems depth with the exception of impacted sites, which show near surface characteristics. Our results suggest that subsurface ecosystems affected by geological degassing are hotspots for microbial life in the deep biosphere.

[1] Environmental Microbiology and Biotechnology, Faculty of Chemistry, University Duisburg-Essen, Essen, Germany. [2] Department of Geology, University Duisburg-Essen, Essen, Germany. [3] Instrumental Analytical Chemistry and Centre for Water and Environmental Research (ZWU), University of Duisburg-Essen, Essen, Germany. [4] Centre of Water and Environmental Research (ZWU), University of Duisburg-Essen, Universitätsstraße 5, Essen, Germany. [5] Geyser-Center, Andernach, Germany. [6] Institute of Applied Geosciences, Karlsruhe Institute of Technology, Karlsruhe, Germany. [7] Present address: Centre for Ecology and Evolution in Microbial Model Systems (EEMiS), Department of Biology and Environmental Science, Linneaus University, Kalmar, Sweden. ✉email: alexander.probst@uni-due.de

The continental subsurface is a huge reservoir for life, hosting about 60% of all microorganisms on Earth[1,2]. Carbon, nitrogen, and sulfur turnover by these microorganisms have a vast contribution to all biogeochemical cycles on the planet[3]. In addition to the great number of microorganisms, subsurface ecosystems can accommodate a large diversity of different bacteria and archaea[4–6], with even single ecosystems containing representatives of almost all known bacterial phyla[4]. Subsurface ecosystems are categorized as either detrital or productive, depending on whether buried organic carbon or inorganic carbon are the main carbon sources of the community[7]. Since no light is available as an energy source in the deep biosphere, alternative electron donors to water like hydrogen ($H_2$) or sulfide ($H_2S$) are used to fuel mostly anaerobic carbon fixation pathways such as the Wood–Ljungdahl pathway[7]. Subsurface lithoautotrophic microbial communities[8] have been reported for many terrestrial ecosystems including the Fennoscandian Shield[9], the Columbia River Basalt[8], the Witwatersrand Basin[10], and subsurface fluids discharged by Crystal Geyser[11]. While these subsurface ecosystems are usually dominated by bacteria, one exception are archaea belonging to the Alti-1 clade of the *Ca.* Altiarchaeota[5,12,13]. Alti-1 form biofilms using their characteristic nano-grappling hooks (hami)[14,15]. The other clade, Alti-2, is more widespread and diverse but found at lower abundances in their ecosystems[14]. *Ca.* Altiarchaeota live autotrophically using the Wood-Ljungdahl carbon fixation pathway[16], which was the most dominant carbon fixation pathway prior to the evolution of photosynthesis[17,18].

Chemolithoautotrophic life in subsurface ecosystems necessitates the presence of adequate electron donors like hydrogen, hydrogen sulfide, or methane. One source of such gases can be Earth's mantle, which also releases $38.7 \pm 2.9$ Tg/yr of oxidized carbon[19], mainly in form of carbon dioxide ($CO_2$), into the crust and the atmosphere[20,21]. This process, also termed mantle degassing, is the transition of volatiles from the mantle (supercritical) to the subcritical zone of the upper crust fueled by lower pressure of volatiles near the surface compared to the mantle[22]. Modern Earth has few areas with active mantle degassing, which are usually restricted to terrestrial volcanoes, subduction zones, or hydrothermal vents in oceans[23–27]. At hydrothermal vents, chemolithoautotrophs initiate the microbial trophic network and proliferate at high rates leading to high microbial cell numbers[1,18,19]. While volcanic sites and hydrothermal vent fields have been studied fairly thoroughly regarding both their microbial community composition and activity[28–32], little is known about deep subsurface ecosystems with low temperatures (283–293 K) and still impacted by gases released from the mantle.

Previous studies have analyzed the influence of mantle degassing via volcanic mofettes, i.e., $CO_2$ seeps below 373 K, on near-surface biomes, particularly soil microbial communities[33–36]. Mehlborn et al.[34] showed that gases from the mantle can alter the availability of different heavy metals including metalloid arsenic and predicted impacts on microbial communities. Beulig and co-workers reported an increase in dark carbon fixation and found evidence that the $CO_2$ from the degassing is indeed incorporated into biomass-based on IR-GC/MS measurements of fatty acid methyl-esters and DNA stable-isotope probing experiments of microcosms fed with $^{13}C$-labeled $CO_2$[35,36]. Along with fermentation processes, the pathways for the turnover of organic carbon were similar in both systems, while the microbial diversity of soils in mofettes was lower compared to controls. Carbon and sulfate respiration were enriched during degassing, while aerobic respiration declined[36] and acetogenesis were suggested to play a major role in these systems[35]. However, these studies were limited to the upper 50-cm of Earth's critical zone, and the influence of mantle degassing on mesophilic microbial communities in the deep subsurface including their metabolic capacity and activity has not been investigated so far.

The cold-water (291 K) Geyser Andernach is located in the Rhine Valley near Koblenz in western Germany and is driven by gases discharged from the mantle[20]. Since 2001 the geyser has had intact tubing, thus tapping into a unique ecosystem. Once released by a mechanical shutter, the gases from the mantle (mainly $CO_2$) permeating the groundwater cause the eruption of cold subsurface fluids sourced from a uniform aquifer system. Thus, Geyser Andernach is an ideal ecosystem to investigate how mantle degassing shapes mesophilic microbial life in the subsurface.

Here, we used a combination of long-term geochemical characterization coupled to genome-resolved metagenomics to investigate the geyser's microbial community. To analyze how mantle degassing impacts mesophilic microbial communities, we set the bacterial replication index values, minimal generation times, and microbial metabolism abundances in Geyser Andernach into relation to 16 other deep continental subsurface ecosystems across the globe. We identified a pattern of decreasing replication indices but shorter minimal generation times with increasing depth. Sites impacted by mantle degassing showed similar replication indices and generation times as near-surface sites, rendering them hotspots for microbial activity in the subsurface. Comparative genomics applied to a key player at sites impacted by geological degassing (*Ca.* Altiarchaeum sp.), revealed that the slow evolutionary rate present in this phylum might be counteracted by horizontal gene transfer (HGT) and gene loss events in this organism group.

## Results

**Geyser Andernach provides access to a stable ecosystem impacted by mantle degassing.** Geyser Andernach was drilled to a depth of 351 m in 1903 tapping into a shale-hosted aquifer with quartz veins. Its eruptions are driven by mantle degassing and can be controlled via mechanical shutters (a diagram of the plumbing system is provided in Supplementary Fig. 1). Geochemical measurements averaged over 14 years have demonstrated that the subsurface fluids provide a constant environment (Supplementary Table 1). The gaseous and ionic composition of the geyser showed the predominance of $CO_2$ in the system and previously reported traces of hydrogen and hydrogen sulfide[20]. Prominent electron donors and acceptors were determined to be hydrogen and ferric iron as well as sulfate, respectively. To investigate the microbial community in subsurface fluids impacted by mantle degassing, we sampled two eruptions of Geyser Andernach and collected the planktonic fraction of microorganisms onto three individual 0.1-µm filters. Metagenomic sequencing of the community resulted in ~7 billion bp per sample (5% SD), covering about 80% of the microbial diversity as estimated by Nonpareil3[37] (Supplementary Fig. 2). Reads were assembled into 921,520 scaffolds on average (20% SD, for further statistics please see Supplementary Table 2). Approximately 75% of the reads (2.6% SD) mapped back the assembly providing evidence that the reconstructed metagenome is representative of the planktonic community at the time of sampling. The community composition based on ribosomal protein S3 (*rpS3*) sequences assembled from the metagenome displayed a fairly restricted diversity consisting of 52 organisms, which spanned twelve phyla (Fig. 1). The core community was composed of 15 organisms detected via *rpS3* across all three metagenomes (Fig. 1), and they accounted for 42.8% (1.3% SD) of the total relative abundance of the community. For 20 of these 52 microorganisms, we reconstructed high-quality genomes with at least 70% estimated completeness (and less than 10% estimated contamination, details in Supplementary Data 1). The most

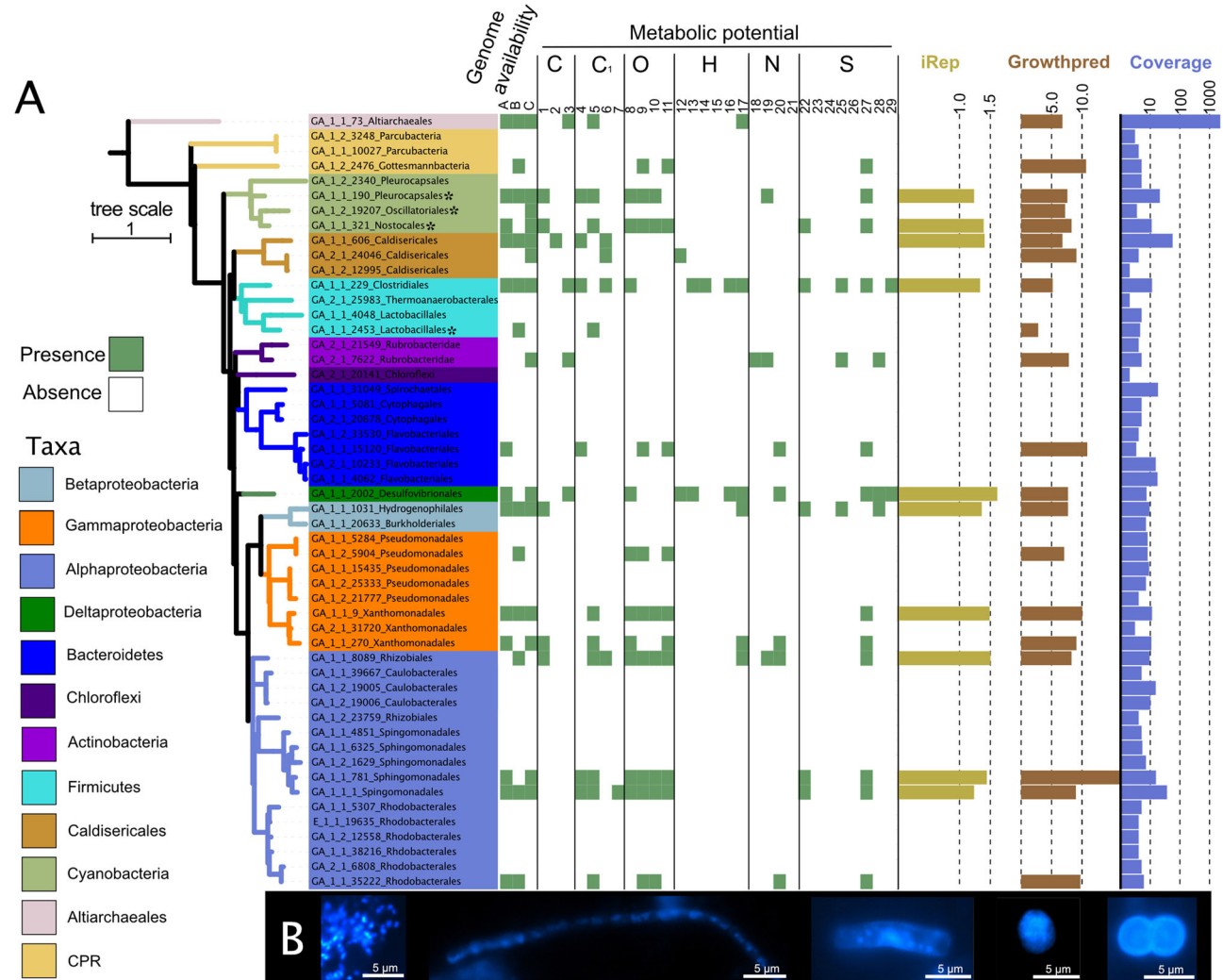

**Fig. 1 Metagenomic and microscopic characterization of the community in subsurface fluids discharged by Geyser Andernach. A** RpS3-based phylogenetic diversity of the organisms in the Geyser Andernach. Centroid rpS3 sequences (after clustering at 99% similarity using cdhit) were used for the calculation of the phylogenetic tree using IQTree. The colors of the different branches signify different phyla. Matching recovered draft genomes in each sample (A–C for samples GA_E1-1, GA_E1-2, and GA_E2-1, respectively), i.e., genomes binned from these samples, are provided as green boxes (otherwise left white). The presence of marker genes based on a marker gene search using HMMs on these genomes for specific chemolithoautotrophic pathways is shown as green boxes (otherwise left white). C signifies carbon fixation with (1) CBB, (2) rTCA, and (3) WL, $C_1$ for $C_1$-metabolism with (4) carbon monoxide oxidation, (5) formaldehyde oxidation, and (6) methanol oxidation, O for oxygen metabolism with (7) cytochrome c bd, (8) cytochrome c bo, (9) cytochrome c $caa_3$, and (10) cytochrome $cbb_3$, H for hydrogen metabolism with (12) FeFe-Hydrogenases type A, (13) NiFe-Hydrogenases type 3b, (14) NiFe-Hydrogenases type 3c, (15) Nife-Hydrogenases, (16) NiFe-Hydrogenases type 4 and (17) NiFe-Hydrogenases type 1, N for nitrogen metabolism with (18) Nitrate reduction, (19) Nitric oxide reduction, (20) nitrite reduction and (21) nitrous oxide reduction, S for sulfur metabolism with (22) sulfide oxidation, (23) sulfite reduction with dsr, (24) sulfite reduction with asr, (25) sulfur oxidation with dsr, (26) sulfur oxidation with sor, (27) sulfur oxidation with sdo, (28) sulfate reduction via APS with sat and (29) Thiosulfate disproportionation. Olive bars show the average iRep value of the respective bacterial population, brown bars show the maximal growth rate of the representative genome as estimated by growth red, and blue bars show the average log10-scaled coverage. **B** Morphologies of microorganisms as determined via DAPI staining and fluorescence microscopy (scale bars = 5 μm) are shown. The morphologies were documented in two sampling campaigns (June 2016 and February 2018 with three and two samples in technical duplicates, respectively).

abundant species recruited 42.8% (1.3% SD) of the metagenomic reads and belonged to the *Ca.* phylum Altiarchaeota[5] (in the following denoted as *Ca.* Altiarchaeum GA) and specifically grouped within the Alti-1[14] clade. The second most abundant organism was classified as Caldiserica, which were originally known to inhabit hot springs[38] but were recently also detected in subsurface ecosystems populated by mesophiles[5,11].

We verified that bacteria in this community were replicating at the time of sampling using in situ replication index values. Replication index values are calculated from the difference of

sequencing coverage between the origin of replication and terminus of replication. Proliferating organisms replicate their genomes with multiple replication forks starting at the replication origin and thus contributing more to sequencing reads. In our study, these index values ranged between 1.4 and 1.5, indicating that 40–50% of those microbial populations, whose iRep values were calculated, underwent genome replication at the time of sampling. Microscopic cell counts of organisms from the subsurface fluids ranged from $2.7 \times 10^6$ to $4.2 \times 10^6$ (average $3.5 \times 10^6$) cells ml$^{-1}$ (Supplementary Fig. 3) and displayed

various morphologies ranging from cocci and rods to filamentous-shaped microorganisms (Fig. 1). Importantly, we also observed clusters of small cocci, which are similar to previously reported biofilm structures of *Ca.* Altiarchaeota[12] and whose presence was confirmed by metagenomic results. We estimated the total amount of erupting carbon ($CO_2$ and hydrogen carbonate($HCO_3^-$)) to be 6270 kg per year, while the microbial cells account for approximately 111.5 g of carbon, suggesting that about 0.0018% of carbon degassing from the mantle is fixed in this ecosystem.

**Replication index values and maximal growth rates across multiple deep continental subsurface ecosystems.** To investigate if mantle degassing has an impact on microbial replication in the continental subsurface, we used in situ replication index values (iRep) of bacterial genomes and maximal growth rate estimates of bacterial and archaeal genomes. We first investigated if iRep can be used as a measure of replication by comparing groundwater fluids to sediments because microbes in sediments are known to be more active[39]. Indeed, iRep suggested a significantly higher replication of microbes in sediments than groundwater ($p$-value < $10^{-3}$). Replication measures from Geyser Andernach were then compared with those from other public datasets from deep subsurface environments of varying depth (overview of samples and ecosystems is provided in Supplementary Table 4). The sampling depth varied from 0 m below ground (cave systems) to 3140 m depth. We reconstructed genomes of previously unbinned metagenomes resulting in 560 newly assembled and classified prokaryotes (Supplementary Data 1) representing 415 different organisms after dereplication. Combined with genomes and iRep results from previous studies[4,5,13], we leveraged in situ replication measures for 895 bacteria (Supplementary Data 2) spanning the vast majority of all known bacterial phyla (see Supplementary Data 5). The average iRep value of bacteria of the individual ecosystems correlated negatively and highly significantly with sample depth across all individual iRep values (Pearson's test, $p$-value < $10^{-8}$) and across median per sampled ecosystem ($p$-value < 0.0007, Fig. 2, Supplementary Table 5). In other words, the deeper the origin of the retrieved sample, the lower the genome replication measure.

In particular, organisms with the capacity of carbon fixation (cor = −0.47), sulfur oxidation (cor = −0.46), or of metabolizing hydrogen (cor = −0.45) contributed to this observation (correlations are summarized in Supplementary Table 6). Samples impacted by high $CO_2$ concentrations, either solely from mantle degassing (this study) or from both mantle degassing and thermal activity[5], were outliers in this correlation analysis. In fact, iRep measures of bacteria in these samples were significantly higher than iRep measures of other subsurface samples ($p$-value < $10^{-15}$) and nearly reached values of samples that are close to Earth's surface (Fig. 2). When excluding these samples from the correlation analysis with depth, the respective correlation coefficient decreased from −0.20 to −0.28 ($p$-value < $10^{-8}$). We also tested how the availability of oxygen influences genome replication measures of bacteria in the continental subsurface. iRep values were on average 0.09 higher for bacteria in oxygenic samples ($p$-value < $10^{-8}$) meaning that about 9% more of the bacteria were undergoing genome replication.

While iRep values indicated that there is less ongoing replication in deeper regions of the subsurface, they do not allow any inference about the speed at which organisms are replicating. Thus, we also calculated maximal possible growth rates, i.e., minimal generation times, based on the codon usage bias between constitutionally expressed ribosomal proteins and the rest of the genes per genome using growthpred[40]. Correlation analyses of

these maximal growth rates with the sampling depth revealed that the maximally possible replication speed increases, i.e., shorter doubling times, with increasing depth ($p$ < 0.0011, cor = −0.143, Supplementary Fig. 4).

**Conserved chemolithoautotrophic metabolism of subsurface microbial communities.** Since bacterial replication is predicted to differ between sites impacted by mantle degassing and reference sets, we investigated if the general metabolism for carbon, nitrogen, and sulfur turnover of entire communities is adapted to high-$CO_2$ subsurface environments. We searched for key enzymes for metabolic pathways across our entire metagenomic assemblies (Supplementary Table 2) and used the abundance of scaffolds that carried a key enzyme as a relative abundance measure of the respective metabolism (Fig. 3, Supplementary Fig. 5, Supplementary Fig. 6). The core metabolism remained relatively stable across all tested ecosystems. We performed both Student's $t$-tests and Kruskal–Wallis tests along with equivalence testing to determine whether there was a significant difference between high-$CO_2$ and non-high-$CO_2$ metabolisms and could only detect a significant difference in the nitrite reduction metabolism (Kruskal–Wallis group comparison, $p$-value = $6 \times 10^{-4}$, details on tests in Supplementary Table 7). Consequently, and in congruence with previous studies investigating the metabolic diversity in a subseafloor aquifer[41], little difference exists in the metabolic potential between regular subsurface microbial communities and those at sites impacted by mantle degassing, although the indigenous organisms at these sites appear to have higher replication index values.

**Biogeography and functional adaptations of deep subsurface Ca. Altiarchaeota.** Key organisms in continental subsurface ecosystems impacted by geological degassing belong to the *Ca.* phylum Altiarchaeota due to their high abundance. *Ca.* Altiarchaeota can currently be divided into two clusters, Alti-1 and Alti-2, with the latter having a broader metabolic variability than Alti-1[14]. In the following, we are going to refer to Alti-1 Altiarchaeota as *Ca.* Altiarchaea. However, organisms of the *Ca.* Altiarchaea is one that can dominate entire ecosystems, as shown for multiple sites across the globe[5,12,13]. Nearly all of the ecosystems are dominated by *Ca.* Altiarchaea has all been reported to have high $CO_2$ partial pressure or great amounts of carbonate deposits[42]. The average nucleotide (ANI) and amino acid (AAI) identity of all so-far recovered *Ca.* Altiarchaea genomes indicated that they belong to the same genus (Supplementary Fig. 7), although 16S ribosomal RNA gene similarity suggested the same species. When correlating the genomic differences based on ANI to the geographical distance between sampling sites of the *Ca.* Altiarchaea genomes, a highly significant negative correlation (*Pearson*, cor = −0.77, $p = 9 \times 10^{-4}$) could be observed, indicating that a greater distance led to greater dissimilarity (Supplementary Fig. 7). We challenged this observation by using robust phylogenetic analyses based on a supermatrix of 30 ribosomal proteins and found that *Ca.* Altiarchaea cluster based on geographical sampling site going all the way to continent-scale (Fig. 4C, Supplementary Fig. 8). However, we did not observe any biogeographic pattern for *Ca.* Altiarchaeota of the Alti-2 clade, which mainly occurs in ocean sediments[14]. Based on Hidden Markov Model (HMM) profiles of key chemolithoautotrophic genes of Alti-2 and Alti-1 genomes, some of which we newly reconstructed from public datasets, we identified substantial differences particularly in the hydrogen metabolism (Fig. 4B, details on *Ca.* Altiarchaeota genomes in Supplementary Table 3). However, Alti-2 showed a significantly smaller minimal generation time than Alti-1 (*U*-test $p$ < 0.0024; Supplementary Fig. 9).

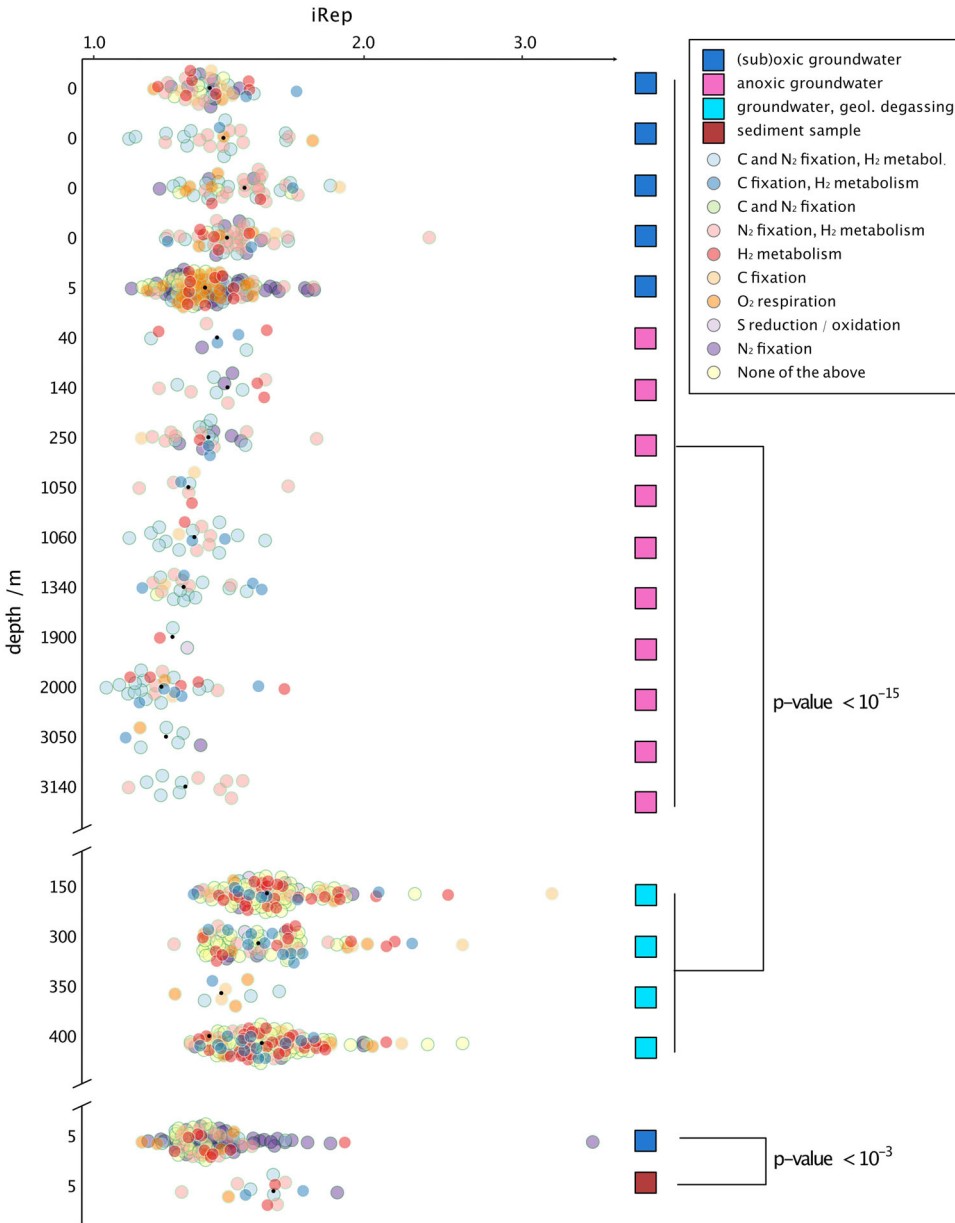

**Fig. 2 In situ bacterial replication rates across subsurface ecosystems ordered by ecosystem depth.** The figure depicts a beeswarm plot of iRep values of genomes (*x*-axis) across ecosystems (*y*-axis) with genomes colored according to their predicted metabolic potential and the black dot representing the median iRep value (individual iRep values in Supplementary Data 2). C represents carbon, $N_2$ nitrogen, $H_2$ hydrogen, $O_2$ oxygen, and S sulfur. Colored squares depict the sample type. Samples impacted by geological degassing and a sediment sample along with the respective aquifer sample are plotted separately. The top *y*-axis shows the sampling depth of the different ecosystems (Supplementary Table 5). In total, 895 genomes were used for this analysis with ≥ 70% completeness and ≤ 10% contamination based on 51 bacterial and 38 archaeal single-copy genes. The order of samples is given in Supplementary Table 5. *p*-Values are derived from two-sided student's *t*-tests. The exact *p*-values from top to bottom are $p < 2.2 \times 10^{16}$ (minimal value in R) and $p = 0.0003934$, respectively.

Since *Ca.* Altiarchaea showed a strict biogeographic pattern, we further investigated their differences in metabolic capacities in depth using a genome model published previously[12] (Fig. 5). We identified that all *Ca.* Altiarchaea share a central NAD(P)H-based Wood–Ljungdahl pathway for carbon fixation and carbon monoxide utilization. The main difference of *Ca.* Altiarchaeum GA to the reference genome *Ca.* Altiarchaeum hamiconexum[12] was the presence of genes for a NiFe hydrogenase (Fig. 4B), which seems to be a specific adaptation to hydrogen-containing gases from the mantle. Indeed, we identified that this NiFe hydrogenase existed in multiple other *Ca.* Altiarchaea and was lost in *Ca.* Altiarchaeum

hamiconexum from IMS. The phylogenetic relatedness revealed that NiFe-hydrogenases of Alti-1 were sister to those of Alti-2 suggesting a conservation of this key enzyme in their last common ancestor (tree is provided in Supplementary Data 7). Other genes are affected by gene loss across *Ca.* Altiarchaea encoded for proteins, which function as mechanosensitive channels, desulfoferredoxin, polysaccharide biosynthesis enzymes, and some peptidases and glycosylhydrolases (Supplementary Data 8–15). By contrast, rubyerythrine and multiple peptidases spanning the families C44 (precursor of amidophosphoribosyltransferase), M06 (metalloendopeptidases), and C01b (endo- and exo-peptidases) were horizontally

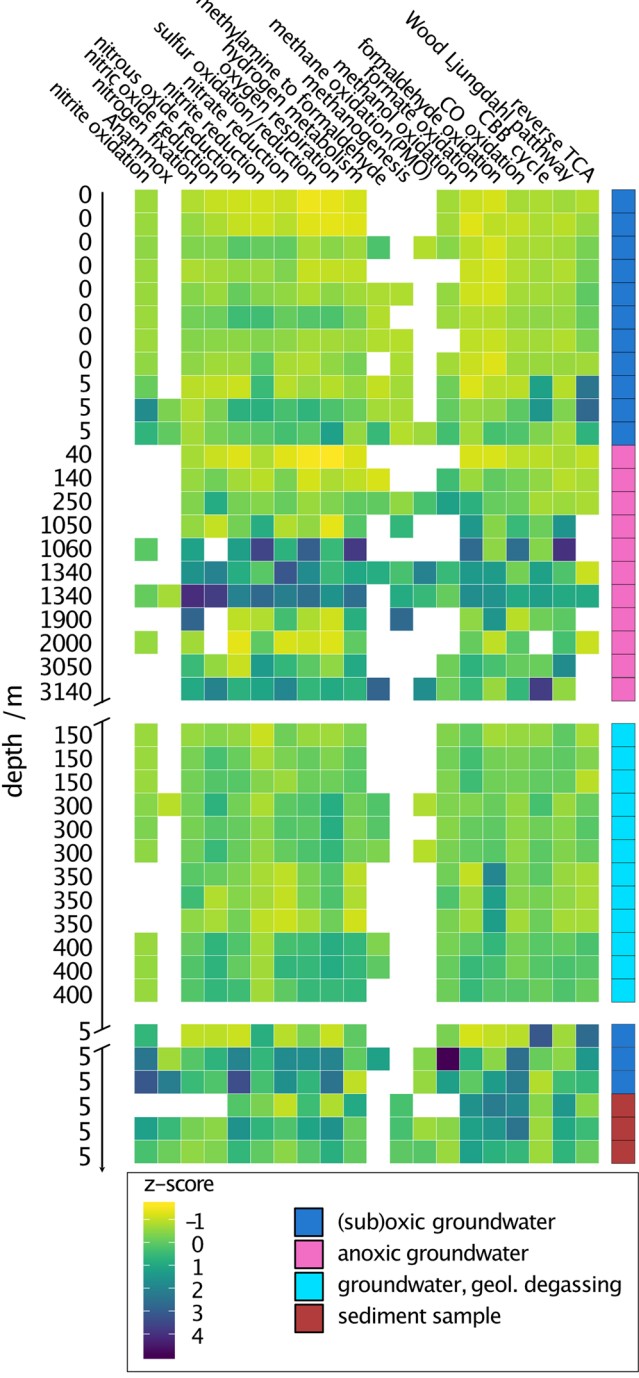

**Fig. 3 Chemolithoautotrophic metabolic potential across ecosystems.** The heatmap shows the read-normalized abundance of chemolithoautotrophic pathways, *Z*-score scaled for the respective metabolisms. Colored squares on the right depict the sample type. If multiple biological replicates of samples were available, up to three were depicted. Sample order is according to Supplementary Table 5. Supplementary Fig. 5 and Supplementary Fig. 6 display the *Z*-scaled number of hits (Supplementary Fig. 5) or normalized abundance (Supplementary Fig. 6) of the individual genes aggregated into their pathways in this figure.

acquired by *Ca*. Altiarchaea species, mostly from the bacterial domain (Supplementary Data 16–19).

This indicates an extreme degree of biogeographic provincialism across Earth. The small genetic divergence of *Ca*. Altiarchaea organisms in their core genome combined with their previously

determined constant cell division[12] implies a very slow evolutionary rate of these organisms. However, gene loss and HGT in *Ca*. Altiarchaea suggests compensation for these slow evolutionary rates potentially providing a substantial advantage over other organisms in deep subsurface environments.

## Discussion

Modeling of current cell counts estimates the number of prokaryotic microorganisms in the continental subsurface to 2 to $6 \times 10^{29}$,[1] which amounts to 60% of the prokaryotic life on our planet[2]. The diversity of microorganisms declines with sampling depth in the continental subsurface[1]. Our metagenome assemblies showed the same trend in diversity change (based on the *rpS3* marker gene, cor = −0.40, *p*-value = 0.021, Supplementary Fig. 10). This indicates that they are representative of general subsurface microbial communities and were consequently used to establish a genome database to calculate genome replication index values and minimal generation times across various subsurface ecosystems. These metrics revealed an apparent contradiction, with both replication index values and minimal generation times decreasing, thus indicating that organisms in the deep biosphere can replicate faster though they replicated less at the time of sampling. Prior studies[43,44] observed a reduction in microbial load with marine sediment depth and age, indicating that communities in older sediments were probably formed by members of surface communities that have a higher degree of persistence compared to others. Thus, subsurface communities would not be formed by actively replicating organisms but instead be shaped by the differing mortality of surface community members[43,44]. The upper ten centimeters of sediment were found to be an exception showing active proliferation[45]. Although we analyzed many different ecosystems, our data do not allow drawing conclusions about the impact of mortality shaping subsurface microbial communities as they originate from different geologic formations. However, our observed decrease in replication measures with sampling depth does agree with these prior observations of a reduction of microbial load with depth and indicates that replication is occurring, albeit with fewer replication forks in the subsurface compared to near-surface ecosystems. On the other hand, the genome structures indicated a faster ability to replicate for organisms in the deep subsurface. This faster possible generation times with depth can be explained by the strategy employed by subsurface microorganisms recently termed as "halt and catch fire"[46]. This strategy refers to an adaptation to nutrient-poor environments like the deep subsurface, where organisms need to adapt to utilize short bursts of available nutrients and thus replicate fast during times when nutrients are available. Sites impacted by geological degassing showed a similar pattern compared to surface samples, both in terms of replication index values and minimal generation time estimates. This could be caused by the unique geology of sites impacted by geological and thermal degassing. In these fracture-controlled aquifers, which are characterized by solid rock formation-embedded channels, flows can reach up to multiple magnitudes greater speeds than flows in comparable sediment-hosted aquifers. Thus, the availability of reduced mantle gases like $H_2$ and $H_2S$ as microbial electron donors highlights the absence of nutrient bursts and the presence of a continuous nutrient flow similar to biomes on Earth's surface.

At Geyser Andernach, *Ca*. Altiarchaeota of the Alti-1 clade reach high cell densities in the $CO_2$ subsurface ecosystem and represent the main primary producers similar to the other high-$CO_2$ aquifer system Crystal Geyser, which additionally harbors a tremendous amount of bacterial diversity but also taps into three different aquifer ecosystems[5,11]. The predicted higher minimal

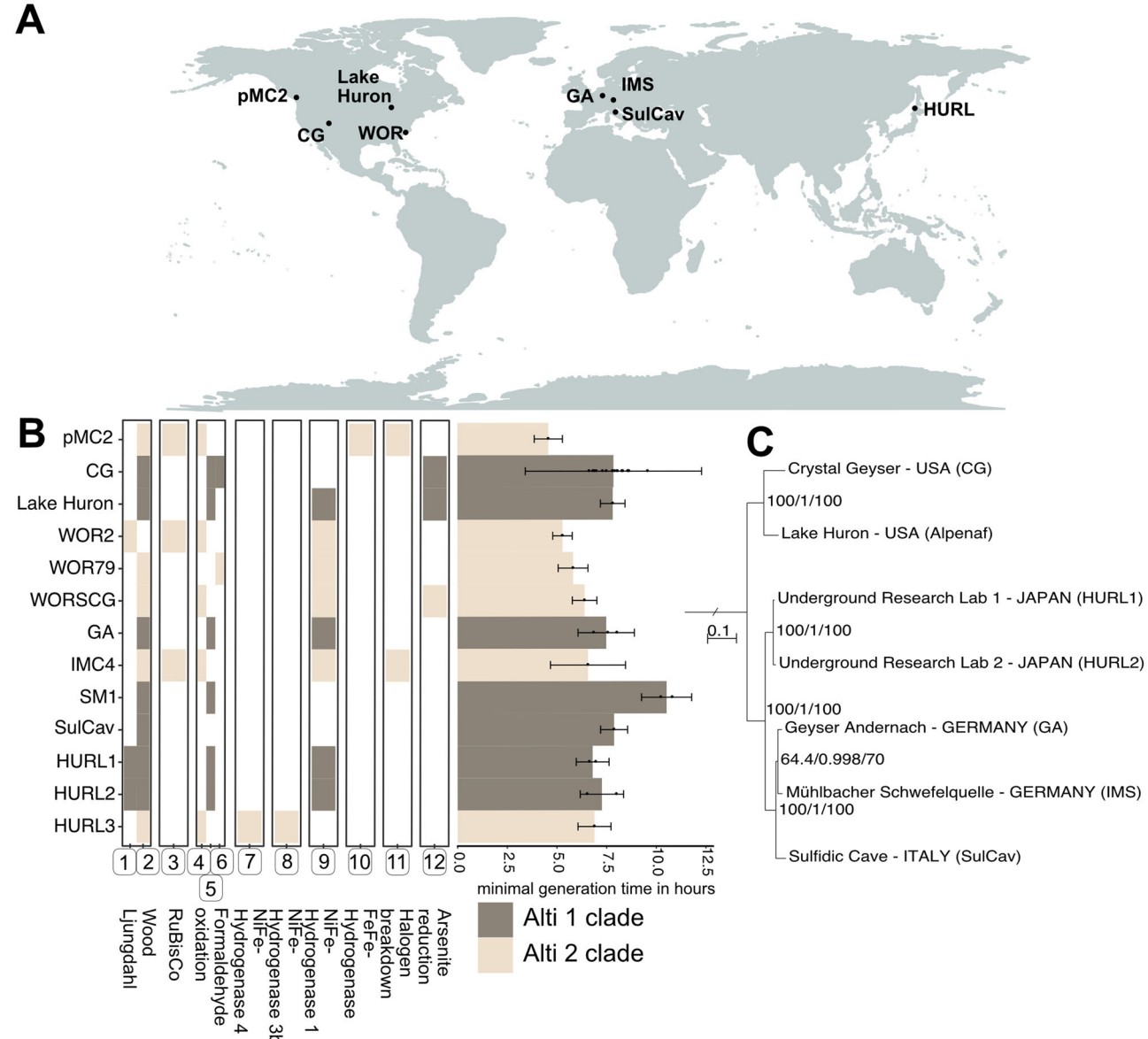

**Fig. 4 Geographical distribution and chemolithoautotrophic potential of *Ca.* Altiarchaeota. A** Global map with locations from which *Ca.* Altiarchaeota genomes were recovered. **B** Metabolic potential of *Ca.* Altiarchaeota genomes. Genomes belonging to the Alti-1 clade are highlighted in dark gray, Alti-2 genomes in beige. If multiple genomes from a specific site were available, they were all used to identify the metabolic potential. The bar chart shows averaged growthpred-predicted minimal generation times across all genotypes recovered from a specific genome, with error bars denoting the averaged standard deviations (growthpred returns both an average minimal generation time and a standard deviation for this value). In addition, the mean minimal generation time for each genome is indicated by black dots. The circled numbers below the heatmap depict the genes identified as markers and stand for (1) codhC, (2) codhD, (3) rubisco form III, (4) fae, (5) fmtf, (6) mtmc, (7) NiFe-Hydrogenase group 4, (8) NiFe-Hydrogenase group 3b, (9) NiFe-Hydrogenase group 1, (10) FeFe-Hydrogenase, (11) hdh, (12) ars. **C** Phylogeny of Alti-1 genotypes based on 30 universal ribosomal proteins (5136 aa positions, IQTree JTTDCMut+F + G4) and using the Alti-2 genome IMC4 as the outgroup. Branch supports correspond to ultrafast bootstraps[77] (1000 replicates), the SH-aLRT test[78] (1000 replicates), and the approximate Bayes test[97], respectively (a tree with outgroup in Supplementary Fig. 8). Details on Altiarchaeales genomes in Supplementary Table 3.

generation time for the Alti-1 clade compared to their sister clade Alti-2 is likely caused by their higher costs of living. In contrast to their sister clade, *Ca.* Altiarchaea (Alti-1) live in biofilms, likely granting them increased survivability against a multitude of biotic and abiotic factors (see Olsen 2015 for a review on biofilm resistance[47]). But this increased resistance also comes with a cost of requiring the synthesis of hundreds of their characteristic cell surface appendages called hami[12,15] as well as other materials making up the extracellular polymeric substances matrix. In addition, *Ca.* Altiarchaea all need to assimilate $CO_2$ via the Wood–Ljiungdahl pathway instead of also supplementing their carbon compounds by taking up organic carbon compounds as only gases can freely penetrate the biofilms. Thus, their proliferation would presumably be much more expensive than for their planktonic sister clade. This leads to the hypothesis that not replication speed but energy requirements limit *Ca.* Altiarchaea proliferation, making an optimization of the codon code to increase replication speed unnecessary.

The abovementioned hypothesis regarding the replication speed of *Ca.* Altiarchaea would also align well with their strict

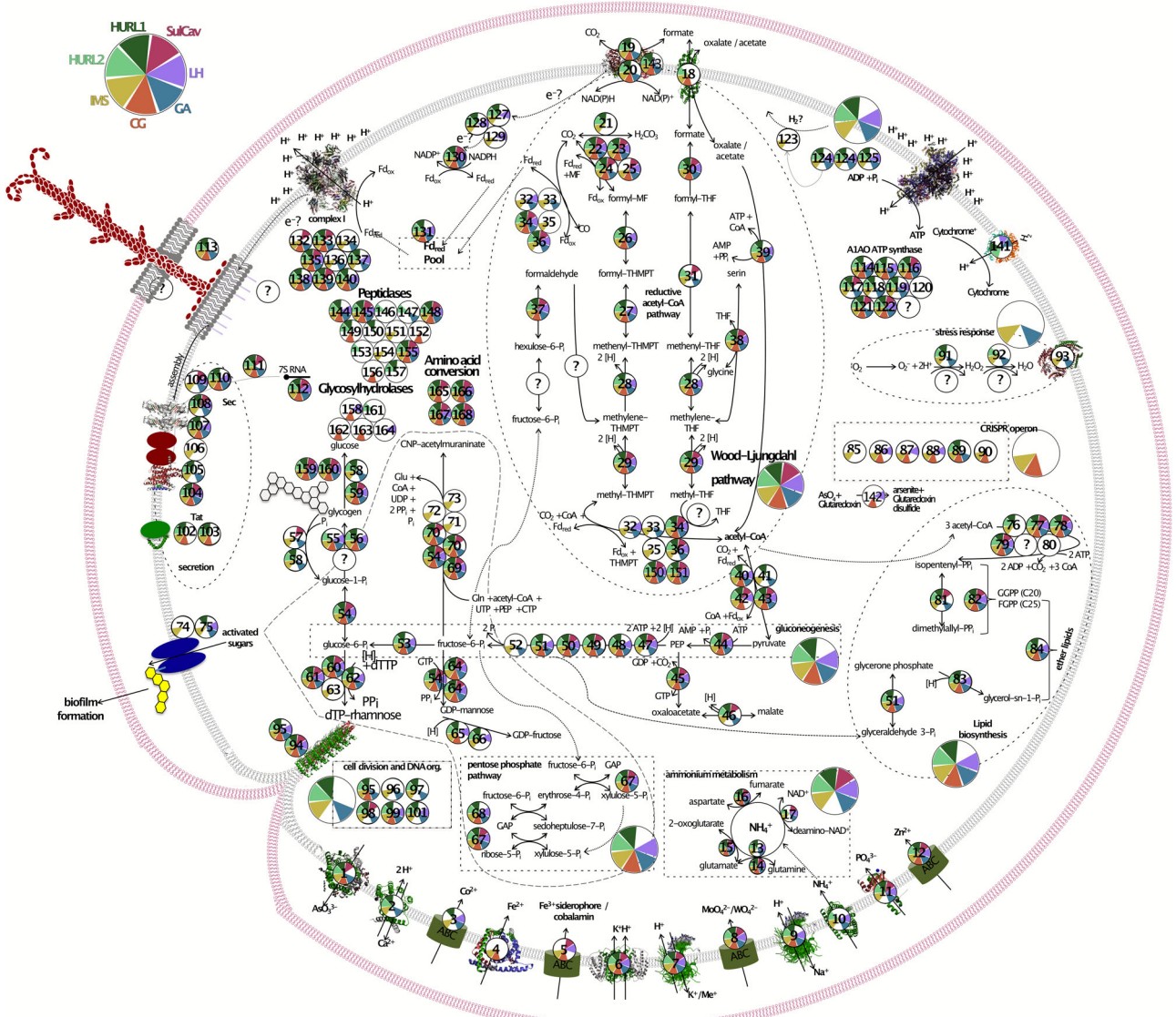

**Fig. 5 Metabolic capacities of *Ca.* Altiarchaeum pangenome.** Previously identified genes in *Ca.* Altiarchaeum hamiconexum IMS[12] was used as the basis to query the other genomes of known Altiarchaea clade members (see Fig. 4 for all members used in this analysis). To expand the predictable metabolic capacity of the genomes, METABOLIC[86] was used to annotate genes, which mainly resulted in peptidases and glycosylhydrolases. If multiple genomes copies per site were available, they were all used to query for the respective genes. All gene functions are listed in Supplementary Data 3.

biogeography. The clustering by continent of origin (North America, Europe, Asia), also reproducible in ANI and AAI (Supplementary Fig. 7), indicates strict provincialism. As dispersal via the surface is unlikely due to the high oxygen sensitivity of *Ca.* Altiarchaea[12], plate tectonics could have been a viable alternative dispersal route providing ample opportunities for the common ancestor to distribute to North America and Europe. Plate tectonics has recently been implicated as the potential dispersal route for *Ca.* Desulforudis audaxviator to Africa, North America, and Eurasia between 55 and 165 Myr[48]. The dispersal of *Ca.* Altiarchaea could have occurred within the Phanerozoic, starting with the early Devonian (~400 Myr), when the continental margins Laurentia and Baltica, which form today's North America and Europe, respectively, collided to form Laurasia[49,50]. Japan, on the other hand, has not been in contact with those margins since the break-up of Rodinia 750–600 Myr ago[51], thus making dispersal to Japan during the Phanerozoic unlikely. As European and Japanese *Ca.* Altiarchaea is indicated to have a common ancestor, one possible route of dispersal from Europe to

Japan could be across the Siberian plate through China in the early Mesozoic and then transferal to Japan during the plate processes, which uplifted the Japanese islands from the sea 25 Myr ago. Future studies are necessary to recover *Ca.* Altiarchaea genomes from Asia further underpin this hypothesis of dispersal since current public datasets from this continent are substantially underrepresented in databases.

The strict biogeography of the *Ca.* Altiarchaea is reflected by the conserved core metabolism, with most pathways being present in every *Ca.* Altiarchaea genome and indicate a slow evolving genus. However, observed putative gene loss and gene transfer events in investigated *Ca.* Altiarchaea populations indicate a compensatory strategy to counteract the slow evolutionary rate. This observed gene loss and transfer might be exuberated by the exclusive living in biofilms, which have generally been known as hotspots of HGT for Bacteria[52]. The genes in *Ca.* Altiarchaea acquired via HGT are mainly from the bacterial domain, an evolutionary process frequently occurring in nature[53]. This HGT likely took place in the subsurface due to the immobility of *Ca.*

Altiarchaea is mediated by the anchoring of cells via their hami. Consequently, our analyses provide evidence that subsurface ecosystems impacted by geological degassing can be hotspots of microbial life and of increased evolutionary rates bolstered by lateral gene transfer across domains.

## Methods

**Geological setting**. The cold-water Geyser Andernach is located 2 km downstream of Andernach (Rhine kilometer 615) on a 0.21 km² peninsula called Namedyer Werth in the Middle Rhine valley. Driven by magmatic $CO_2$, the geyser erupts regularly and intermittently approx. every two hours, when the groundwater filling the well is saturated with $CO_2$ and a reinforced chain reaction (domino effect) concludes in a gas/water-eruption up to >60 m in height[54], lasting for 15–20 min. The well (drilling Ø 750/312/216 mm; casing/screens Ø 150 mm) was drilled in 2001 and is the third borehole (after 1903 and 1955) on this peninsula. The drilling taps 14 m of Quaternary fluvial deposits and continues then until its total depth of 351.5 m in a lower Devonian formation called "Hunsrück Schiefer s.l." (shale)[55]. A diagram of the plumbing system of the Geyser Andernach is provided in Supplementary Fig. 1.

The small peninsula is part of the Pleistocene terrace which is covered by a thin sandy layer of fluvial Holocene deposits. Only at the NE margin of the peninsula, the terrace is bare of deposits. The thickness of the Quaternary layer varies from 14 m (drilling 2001) to 20.75 m (drilling 1903)[56] and 24.2 m (drilling 1955) in the vicinity of the cold-water geyser. Beneath the Quaternary deposits follow lower Devonian rock formations of low metamorphic shale, such as clayish shale and intercalated minor layers of quarzitic sandstones; the thickness of these series is up to 5000 m.

The peninsula is located in the Middle Rhine Valley, which is a part of the European Cenozoic Rift System[57]. This rift system runs between the cities Bingen and Bonn in SE–NW-direction and crosses the Variscan complex of the Rhenish Massif. Located at the SE edge of the lower Middle Rhine Valley, Geyser Andernach is situated on the intersection of two major fault structures: about one km to the NW the Variscian Siegen thrust fault running SW-NE crosses the Rhine Valley and can be traced for over 100 km from the Eifel area to the Westerwald. This fault shows a vertical displacement of several thousand meters, which occurred during the Variscan orogenesis, thus bringing rocks of the middle Siegenian stage in lateral contact with the lower Emsian stage[58]. About 2 km to the SE the lower Middle Rhine valley is morphologically separated from the adjacent intraplate Tertiary Neuwied basin by an approx. 100 m vertical displacement caused by the SW–NE trending Andernach fault.

The Andernach fault and the Siegen thrust fault were in post-Variscan time intersected and 200–300 m displaced by a SE–NW trending dextral strike-slip fault[59,60]. The fault is supposed in the river Rhine bed and covered by Quaternary deposits. The horizontal movement was probably combined with shear strain and cataclastic rocks in the vicinity of the fault. This fault is the cause for pathways of mantle gases to reach the subsurface aquifers and ultimately the atmosphere.

Starting in the Tertiary, a mantle plume under the Eifel area caused an uplift of the Rhenish massif during the last two million years and is the driving force for the volcanic activity in the Quaternary Eifel area since 700 k years[61].

The mantle plume is the basic requirement for the rise of magma under and into the crust, whereby magmatic gases are released.

**Sampling and geochemical measurements**. The mesophilic and $CO_2$-driven Geyser Andernach (50.448588°N, 7.375355°E) in western Germany was sampled on 21 February in 2018 by a collection of erupting water in sterile, DNA-free containers and subsequent filtration onto 0.1 μm pore size filters of 142 mm diameter (Merck Millipore, JVWP14225) and storage on dry ice/193 K until DNA extraction. Water samples were collected during the eruption of the geyser and analyzed biochemically as well as microscopically (see Supplementary material for details). In total, two sequential eruptions were sampled, resulting in two filter samples for the first eruption and one filter for the second eruption. The upper 83 m of the geyser will have a casing and are sealed with cement so that no water can enter the well from the sides. The residual length of the geyser borehole (83–351.5 m) is intermittently covered by bridge-slotted screens which allow entry of $CO_2$-saturated water into the geyser well (Supplementary Fig. 1). Each eruption flushes the tubing system (cylindric shape, 7.5 cm radius, 351.5 m length, approximate volume 6.2 m³) with 6–7 m³ water and an additional eruption was performed prior to the sampled eruptions to rid the tubing system of any stagnant water. The metagenomes recovered from both eruptions show identical community compositions and consequently, the sampled communities should be representative of subsurface communities and not contamination from the tubing system.

**Metagenomic sequencing and processing**. DNA was extracted from three individual 0.1 μm bulk water filtration filter membranes using the DNeasy PowerMax Soil DNA Extraction Kit (Qiagen, JVWP14225) according to the manufacturer's instructions and further concentrated using ethanol precipitation with glycogen as the carrier. The samples were sequenced as part of the Census of Deep Life phase 13 sequencing grant using Illumina NextSeq (paired-end, 150 bps each). The three samples were processed individually as follows: Quality control of raw reads was performed using BBduk (Bushnell, https://sourceforge.net/projects/bbtools/) and Sickle[62]. The metagenomic coverage and sequence diversity of metagenomes was estimated using Nonpareil3[37] using k-mers of size 20. Reads were assembled into contigs and scaffolded using metaSPAdes 3.11[63]. For the sample IMS-BF, a sub-assembly of reads not mapping to the available Ca. Altiarchaeum SM1 genome (GCA_000821205.1) was performed to improve assembly quality and this sub-assembly was used for the binning of additional genomes. Open reading frames were predicted for scaffolds larger than 1kbp using Prodigal[64] in meta mode and annotated using DIAMOND blast[65] against UniRef100 (state Dec. 2017)[66], which contained the NCBI taxonomic information of the respective protein sequences. The taxonomy of each scaffold was predicted by considering the taxonomic rank of each protein on the scaffold on each taxonomic level and choosing the lowest taxonomic rank when more than 50% of the protein taxonomies agree. Reads were mapped to scaffolds using Bowtie2[67] and the average scaffold coverage was estimated along with scaffolds' length and GC content.

**Binning of GA samples**. Abawaca[68], MaxBin2[69], tetranucleotide-based Emergent Self-Organizing Maps (ESOM[70]), and CONCOCT[71] were used to identify metagenome-assembled genomes and DAS Tool with standard parameters was used to aggregate the results[72] (see Supplementary Methods for a detailed listing of the parameters used). Binning of publicly available datasets was carried out using a combination of MaxBin2, Abawaca, and tetranucleotide ESOM, if possible. Bins were refined using GC content, coverage, and taxonomy, and their completeness and contamination were accessed by a set of 51 bacterial and 38 archaeal single-copy genes as described previously[5,11]. Only bins with ≥70% estimated completeness and ≤10% estimated contamination were used for downstream analysis. For each sample, genomes were dereplicated using dRep[73].

**Ribosomal protein S3 (rpS3) analysis**. Genes annotated as ribosomal protein S3 were extracted and assigned to genomes where possible based on shared GC, coverage, and taxonomy. rpS3 coverage was determined based on the scaffold coverage (see above) containing the ribosomal protein. Ribosomal protein sequences were clustered using MUltiple Sequence Comparison by Log-Expectation (MUSCLE)[74], trimmed using BMGE 1.0[75] with the BLOSUM62 scoring matrix, and aligned using IQ-TREE[76] multicore 1.3.11.1 with -m TEST -bb[77] 1000 and -alrt[78] 1000 options. The tree was visualized along with other genomic data using the iToL platform version 5.5[79].

**Identification of potential contaminant genomes**. The GTDB-Tk[80] classify_wf workflow with default parameters was used to place the recovered genomes from the Geyser Andernach in relation to a reference dataset. If a close relative genome was identified in this approach, we calculated the ANI between the reference and the newly recovered genome. The only genome showing a similarity ≥80% ANI to the reference dataset was GA_180221_E-1–2_metaspades_Carnobacterium_36_4 (96.42% ANI to *Carnobacterium alterfunditum* GCF_000744115.1) and was thus identified as a potential contaminant and excluded from further analyses.

**Determination of bacterial in situ replication index**. Reads were mapped onto concatenated genomes per sampling site using Bowtie2 with the reorder flag[67] and the index of replication (iRep[68]) was calculated, allowing for 2% mismatches relative to the read length (3 mismatches for 150 bp). The calculation of in situ replication index values is based on the assumption that organisms, that are actively proliferating, replicate their genome starting at the origin of replication and ending at the terminus of replication. Replicating organisms can thus have already replicated the parts of their genome close to the origin of replication but have not yet completed replicating sequences close to the terminus of replication. This can result in higher relative coverage of the sequence close to the origin of replication compared to the terminus of replication. Multiple simultaneous replication processes can exuberate this difference further. The in situ iRep estimates the number of replication processes based on this coverage difference but only works in Bacteria as Archaea can have multiple origins of replication[81] and thus the iRep signal is distorted and cannot be applied in a comparative manner. If multiple samples were available for one ecosystem, all iRep values for one genome were calculated and averaged to ensure comparability with other samples.

**Prediction of maximal growth rates**. Growthpred[40] values were calculated on prodigal-predicted genome gene sets in nucleotide format with the -t parameter and otherwise default options. Growth rate estimators like Growthpred utilize differences in codon usage between genes which are continuously expressed like housekeeping genes (by default growthpred uses ribosomal proteins) and the rest of the gene pool to predict how optimized the genome is for a faster replication. In contrast to iRep, growthpred does predict the actual fastest rate at which a genome can replicate.

**Metabolic potential predictions**. A set of HMM with respective score thresholds for chemolithoautotrophic key enzymes[4] was used to predict the metabolic potential of recovered genomes and overall in entire assemblies (see Supplementary material for more detailed information).

**Biogeographical analysis**. The R package sp[82] was used to calculate the geographical elliptical distance between two sampling sites (based on longitude/latitude), in which putative genomes of the *Ca.* Altiarchaeales subclade Alti-1 was identified. The average nucleotide identities (ANI) between all available putative genomes of the *Ca.* Altiarchaeales subclade Alti-1 was calculated using the ANI calculator[83] with default parameters. Correlations between geographical distance and ANI were done using Pearson's *r*[84].

**Genome comparison of *Ca.* Altiarchaeota**. Genes of all *Ca.* Altiarchaeota genomes were blasted against each other (*E*-value: $10^{-5}$) and matches were filtered to matches with the similarity ((alignment length × density)/query length) thresholds of ≥40%, 50%, 60%, 70%, or 80%. Cytoscape 3.7.2[85] was used to visualize the networks at the respective similarity thresholds.

**Metabolic network of Ca. Altiarchaea (Alti-1)**. The annotated genes from Probst et al.[12] were used as the basis to identify homologs in other Alti-1 genomes using an *E*-value of $10^{-5}$ as the cutoff. If multiple versions of a genome were available, their results were concatenated. In addition, genomes were annotated using METABOLIC[86], mainly incorporating annotations for glycosyl hydrolases, peptidases, and aminotransferases.

**Phylogenomic analysis of *Ca.* Altiarchaeota**. Amino acid sequences and annotations for Alti-1 ORFs plus one Alti-2 serving as outgroup were predicted using Prokka 1.14.0[87] with options: --kingdom archaea --metagenome --compliant. The resulting protein datasets were searched with HMMER 3.2.1[88] for homologs of 30 universal ribosomal proteins using the v4 HMM profiles from Phylosift[89]. A $10^{-4}$ cutoff was applied, and the resulting datasets were curated manually to remove distant homologs and multiple copies in each genome, as well as to fuse contiguous fragmented genes. Individual genes were aligned with MUSCLE v3.8.31[74] and trimmed with BMGE[75] under the BLOSUM30 matrix. The genes were then concatenated into a supermatrix of 5156 aa positions. The phylogeny was reconstructed in IQTree 1.6.11[76] under the JTTDCMut+F + G4 model as selected by ModelFinder[90].

**Tracking of gene loss and gene transfer events in *Ca.* Altiarchaea**. To identify genes that were lost in multiple *Ca.* Altiarchaea or identify genes that were acquired by individual *Ca.* Altiarchaea through HGT, we selected genes only present in one or two *Ca.* Altiarchaea genomes (Fig. 5) for phylogenetic analyses. The selected genes were used as BLASTp queries (*E*-value: $10^{-5}$) against a reference database of bacterial and archaeal genomes, retaining up to 2000 hits per search. The database is a concatenation of bacterial and archaeal genomes in the NCBI Genome database (accessed 2019.06.01), dereplicated using rpS3 amino acid sequence clustering with CD-Hit at 99% identity followed by dRep at 95% ANI to get a single representative genome per species. This resulted in a databank of 25,226 bacterial and 1808 archaeal genomes. Taxonomic information and functional annotation (when available for genomes with protein datasets) were used directly from NCBI. If no protein dataset was available, the translated ORFs were predicted with Prodigal. Genes were aligned with MUSCLE, trimmed using BMGE with the BLOSUM30 matrix and their phylogeny was reconstructed using IQTree2.0-rc2 with the -m MFP, -bb 1000, and -alrt 1000 options.

**Community-wide analyses**. Genes were predicted on assemblies with scaffolds longer than 1 kbp and chemolithoautotrophic key enzymes were predicted as described above. The abundance of the genes was estimated using the coverage of the encoding scaffolds after adjustment to unequal sequencing depths by normalization using the total bps per library. If a pathway was represented by multiple key enzymes, the enzyme with the highest frequency of hits was selected. Abundances of individual key enzymes were summed to provide the total relative abundance of each pathway in the respective samples. Likewise, diversity within each assembly was estimated based on *rpS3* diversity and relative abundance of the respective scaffolds.

**Estimations of annual total erupted carbon and intracellular erupted carbon**. The annual total erupted carbon was calculated based on the available $CO_2$, $HCO_{3-}$, and cell concentrations, the eruption volume (Supplementary Table 1), the average estimate of the intracellular carbon amount from Kallmeyer et al.[91] of 14 fg cell$^{-1}$, and the number of eruptions during tourist season (roughly 1 April–31 October ~210 days). See the Supplementary Material for the calculations.

**Statistical analysis**. Statistical analyses were performed in the R programming environment[84]. These included paired and independent *t* tests, Pearson correlations, analysis of variance (ANOVA), TukeyHSD significance tests[92], the

Shannon–Wiener index[93], and equivalence testing using TOSTER[94]. As the upper and lower equivalence boundaries for equivalence testing of two groups, we used the effect size the $CO_2$-poor sample group had a 33% power to detect as recommended previously[95]. Results were visualized using ggplot2[96].

Methods for DAPI staining, cell counting, geochemical measurements are provided in the Supplementary Methods.

## Data availability

Raw sequencing data and MAGs from Geyser Andernach have been deposited at SRA and Genbank, respectively, and are available under the BioProject PRJNA627655. MAGs binned from additional ecosystems have been deposited at Genbank in the BioProject PRJNA767587. Individual BioSample IDs of all MAGs are listed in Supplementary Data 1 and individual SRA accession codes are listed in Supplementary Table 4.

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

## Acknowledgements
We thank Hubert Müller for technical assistance, Sabrina Eisfeld for laboratory maintenance, Ken Dreger for server administration and maintenance, and Karen L. Lloyd for scientific discussions.

## Author contributions
T.L.V.B. performed the main bioinformatics analysis. P.S.A. performed phylogenomics. V.T. and A.J.P. performed microscopy. U.S., R.S., and B.K. performed geological analyses and geological data interpretation. T.L.V.B. and P.A.F.G. analyzed genomes. T.L.V.B., J.R., and A.J.P. took samples. D.K. and T.C.S. performed geochemical analyses. A.J.P. conceptualized the study. T.L.V.B. and A.J.P. wrote the paper with revisions from all co-authors.

## Funding
This study was funded by the Ministerium für Kultur und Wissenschaft des Landes Nordrhein-Westfalen (Nachwuchsgruppe Dr. Alexander Probst). The Geyser Andernach metagenomes were sequenced within the Census of Deep Life Sequencing call 2017, phase 13 project Microbial metabolism in a deep subsurface, shale-hosted aquifer of the Volcanic Eifel (central Europe): a comparative analysis of two cold, high-$CO_2$ geysers. JR received funding by the DFG (RA 3432/1-1) during revisions of the manuscript. Open Access funding enabled and organized by Projekt DEAL.

## Competing interests
The authors declare no competing interests.
