## [Peer Review File · Nature Communications]

REVIEWER COMMENTS

Reviewer #1 (Remarks to the Author):

This study presents metagenomic analysis of three groundwater samples from a CO₂-rich site, to explore if mantle-derived carbon stimulates a subsurface biosphere. This particular environment is enriched in Altiarchaeota, a candidate DPANN phyla without a representative isolate (*Candidatus Altiarchaeum hamiconexum* notwithstanding). The results of this study are compared to similar prior studies from other groundwater systems to examine if there are trends in deep biosphere diversity, growth, and metabolic function as a function of depth, geography, and/or environment type. Considering that continental microbes may comprise 60% of microbial biomass on Earth, this study could be of interest to a wide scientific community.

In my assessment, there are 4 major claims of this study, although they are not necessarily presented with equal weight in the presented manuscript: 1) That bacterial replication (and, by extension, activity) declines with depth, except in areas of mantle-derived C stimulation (mainly in Figure 3); 2) That there is little variation in metabolic potential despite geographic distribution patterns of taxa (Figures 2 and 4); 3) that *Altiarchaeum* from the new study site indicate an adaptation to hydrogen from the mantle by possessing an NiFe hydrogenase (Figure 2); and 4) that Altiarchaeota have a slow evolutionary rate of genetic divergence.

Regarding claim 1, I have concerns about the application of the iRep proxy for supporting this claim. The index of DNA replication (iRep) is a proxy for the instantaneous DNA replication rate of cells, calculated from the ratio of metagenomic read coverage near the origin of replication versus the terminus in individual genomes-from-metagenomes (i.e. MAGs), with assumptions made about the ordering of genomic bins based on coverage. While this approach has been validated with laboratory cultures, there is limited verification in environmental samples, and at least some indication that it is not a robust indicator of replication/growth (for example, Long et al. 2019 Biorxiv DOI: 10.1101/786939). Application of iRep to environmental samples may be complicated by factors like horizontal gene transfer, or close relatives cross-recruiting mapped reads, or extracellular DNA, that impact read recruitment but are not indicators of population ecology. Its application may also be complicated by a lack of understanding of the relationship of replication and growth in slowly growing populations replicate under energy-limiting conditions, as would be the case in the subsurface. Furthermore, the assumption that all cells in each sample are in physiological states representative of in situ conditions will be dependent on sampling strategies, which is not controlled in the meta-analysis (at least, not indicated in Supplemental Table 6). Also problematic to me is that the study only calculated iRep values for 10 of the 52 microbial groups claimed to be in the community (Figure 1); how then can it be asserted that 40-50% of the population was undergoing replication when there are not values for 80% of the population? Thus, while there are clear differences in the iRep values of different samples, and that the samples exposed to mantle-derived carbon have higher values (i.e. Fig 3), I am not convinced that the interpretation of those is as robust as the authors claim (i.e. that it indicates that “geological degassing leads to...higher microbial activity” L34; L285: “This finding validates that replication indices can be used as proxies of activity in

these subsurface ecosystems" L 286: "Using these measures across multiple samples from different depths established that the activity of bacteria declines with sampling depth, suggesting a deep biosphere with little activity"). Because there are possible confounding factors that could influence the iRep value, as indicated above, I think it is speculative to state that these values are indicators of activity. I think that the authors need to more cautiously interpret the variation in iRep values to account for these possible confounding factors, and to considerably tone down the use of "activity" as a replacement for "replication" (since some reviewers, like myself, equate activity with a time-dependent rate).

Beyond these concerns about whether or not the iRep are actually indicating changes in replication with depth, I encourage the authors to put their discussion about changes in the deep biosphere with depth in the perspective of other recent papers on this topic, such as D'Hondt et al. 2019 Nature Communications, Kirkpatrick et al. 2019 Frontiers, Lloyd et al. 2020 Biorxiv, Starnawski et al. 2017 PNAS.

Regarding the 2nd major claim, that there is little variation in metabolic potential despite geographic distribution patterns of taxa (Figures 2 and 4), this is not a surprising outcome to me considering other studies that have documented metabolic redundancy even when diversity fluctuates, including in the subsurface such as Tully et al. 2017 The ISME Journal.

Regarding the 3rd claim, that Altiarchaeum from the new study site indicate an adaption to hydrogen from the mantle by possessing an NiFe hydrogenase (Figure 2), these species-specific differences are mainly addressed in the results. If the authors tone down the iRep-based inferences in the paper, this could open up space for digging deeper into Altiarchaeota-specific biogeography and functional variation within the manuscript.

Claim 4, that Altiarchaeota have a slow evolutionary rate of genetic divergence (for example, L187), isn't brought up in the abstract, but it is sizeable point of discussion in the paper and supplement. I find this discussion interesting, and encourage the authors to pursue it even more. As above, if a revised paper de-emphasizes the iRep-based claims, then this could open up space for presenting a deeper presentation of the Altiarchaeota.

Overall, I found this paper to compress two stories together – one about replication/activity in the subsurface, and the other about Altiarchaeota biogeography. As expressed above, I have reservations about the assumptions and interpretations of the replication-oriented story. By contrast, I think the Altiarchaeota focus story could be further developed. Therefore, I recommend that the authors de-emphasize the replication-based story, and be more conservative in the interpretations based on the caveats of the assumptions of what iRep is actually measuring, and instead shift the focus of the paper to the Altiarchaeota.

Despite my concerns about the interpretations, I do commend the authors on writing a very nice manuscript; it was easy and engaging to read. I caution the authors to carefully review the citation callouts, as there seems to be some mismatches between the numbered references and the actual citations I think they meant to callout.

Overall the methodological details provided were appropriate (except where noted below), although I must flag that the authors did not provide the SRA accession numbers for their metagenomic sequence data (only indicated as "XXX" in Supplemental Table S5, and not mentioned in the text), so it was impossible for the reviewer to evaluate the data unless knowing where to find the data within the Census of Deep Life. This makes reviewing a sequence-based manuscript very difficult, if not impossible, to verify the data processing steps. As a reviewer, I cannot recommend accepting a manuscript that does not provide access to all of the data. Also, it does not appear that the authors included a no-template-control in their sequencing, which is problematic for a low biomass study (Sheik et al. 2018 *Frontiers*).

Minor suggestions:

L19: consider adding additional references to the support the statement that "releases huge amounts of oxidized carbon...into the crust and atmosphere", such as Werner et al. 2019 (<https://www.cambridge.org/core/books/deep-carbon/carbon-dioxide-emissions-from-subaerial-volcanic-regions/F8B4EFAE0DAF5306A8D397C23BF3F0D7>), to support the one field-based study cited.

L72: Consider also adding the Fullerton et al. 2019 study from Costa Rica mantle degassing systems, as another example: <https://eartharxiv.org/gyr7n/>

L119: The manuscript states that 15 organisms were detected in all three metagenomes, but when I look at Figure 1, I only count 7 rows with genomes in each three samples. What am I missing?

L122 (and L240 and L269 and L308): "Interestingly" is subjective (see <https://www.nature.com/articles/ni1105-1061> for more discussion of not using such "helper words" in scientific writing); I suggest removing this word.

L128: Based on Figure 1, you only calculated iRep values for 10 of the 52 microbial groups you claim are in the community. Since you do not have these estimates of this index for ~80% of the community members, I think it is erroneous to say that 40-50% of the population was undergoing replication at the time of sampling. That metric can only be applied to the 20% of taxa that the estimate could be made for. Please rephrase this statement accordingly.

L135: "t" tonne is not an SI unit. I am assuming you mean a metric ton, which is 1000 kg. Please use proper SI units through the manuscript and supplemental text. For example, 6.27×10^3 kg. This will also make the comparison to the calculation of microbial biomass easier for the reader.

L282: I don't think you have the correct reference cited for the statement about higher replication rates in sediment versus plankton? Hoang et al. 2018?

L284: I don't think that Figure S4 is the correct callout to support this statement? Do you mean Figure 3?

L300: The argument about plate tectonic driven dispersal of Alti-1 is highly speculative, so I don't think that "most plausible mechanism" is the correct phrasing here. I suggest "a possible mechanism".

L324: this sentence about Early Earth is HIGHLY speculative. There are so many factors to consider that would influence microbial activity, besides just mantle degassing driven stimulation, that it is a wild guess if there was a "high microbial activity" at the surface. I think that this speculation detracts from the findings and strongly suggest removing it.

L376: Was a no-template control sequenced in this study? Some of the proteobacterial clades contain taxa that are commonly detected as lab contaminants in sequencing efforts that have proper controls (*Pseudomonas*, *Sphingomonas*). The authors should compute ANI values for these bins compared to genomes of cultures / MAGs / SAGs in NCBI/IMG via a tool like FastANI to understand and communicate how likely these bins are contaminants. Any Bin with a suspiciously high ANI value to anything that is not subsurface should be removed and all downstream statistics would need to be recalculated.

L381: It is unclear whether the samples from this current study were "co-assembled" (reads from DNA extracts from all three filters assembled together) or assembled one at a time. Line 58 in supplemental methods uses the phrase "differential coverage binning" which suggests they were co-assembled but the wording isn't clear prior to that.

Figure 1: the text is impossible to read in this two-column figure. Consider making the phylogenetic tree branching portion of the image thinner in horizontal space to allow more room for text. Also, the figure does not have A and B panel labels, although the figure caption indicates an A and B. Furthermore, as presented, lining up the DAPI images vertically suggests that the morphologies are somehow connected to specific phyla, which is not the intention that I think you mean to convey. Consider moving the DAPI images to be in a row below the metagenomic figure panel (which again will provide more space to make the text larger).

Figure S4: This is a weak correlation with a high p-value and it looks like it is really influenced by the higher number of near-surface samples and much lower amount of deeper samples.

Reviewer #2 (Remarks to the Author):

Summary

Bornemann et al. performed metagenomic analysis, along with collecting ancillary geochemistry, on a geyser system. It provides novel information on a cooler endmember of such systems and is contextualized with results from other subsurface studies to interpret the role of carbon fixation utilizing the high CO₂ in these waters. The manuscript contains a lot of work and a lot of novel metagenomic data. However, the current presentation overstates claims and does not provide enough information to allow the reader to evaluate interpretations, as detailed below.

Major Issues

Pathways and genes used: Since it is a major component of the paper, the specific genes used to determine metabolic capabilities should be included in the manuscript rather than just a reference to another paper. The figures should list the specific gene used and there should be a table of which genes were used from each metagenome to determine all of the metabolisms discussed in the paper. This would also add clarity for if only one of the genes was used to represent a given metabolism, or if all were required to declare presence of the metabolism. Were any complete pathways recovered? There are also nuances to the genes used in Anantharaman et al 2016. For example, from your reference's SD14, propionyl-CoA synthase used for 3HP cycle is a gene that is also used in other pathways than 3HP, which is another reason to be explicit in this work which genes were used.

Activity: The interpretation of activity is overstated since only a proxy for activity was used. iRep metrics could be used in conjunction with activity findings by other means (e.g. SIP or metaT), but cannot be used on their own to support the claims of microbial activity in the abstract and throughout. This is especially true given its use on uncomplete genomes, where the metric was only 70% completion or above, and the 5m sediment vs water comparison has enough variability that it does not suffice in place of an actual activity measurement. Therefore, the manuscript should be revised to describe findings as a potential proxy for activity not an actual measurement of activity.

Also as no activity measurements were conducted, metagenomic analysis should be more clearly couched as potential carbon fixation/carbon fixation potential rather than definitive active carbon fixation.

Carbon in cells: 90 fg C/cell is high for a subsurface ecosystem. See Kallmeyer et al. 2012 PNAS for a more recent discussion of the choices involved in choosing a number for this in subsurface systems.

System plumbing and contamination controls: Cell densities are very high for subsurface fluids. While it appears the recovered organisms that are the focus of the paper are environmentally relevant and not common contaminants, Figure 1 contains some groups that are often contaminants (Salter et al). It is important to constrain where in the system the Alt. archaea thought to be located to support the interpretation of their role in carbon cycling. Are there areas where stagnant waters could collect between sampling? Are they flushed for a period of time with some metric to ensure formation waters are being sampled? Are biofilms able to grow along sampling lines and these cell densities are instead measurements of biofilms? A diagram of the mechanics of the sampling system/plumbing with a depiction of where microbial communities are expected to reside within this system and a discussion of how sampling of formation fluids has been verified are required for interpretation of these results.

Minor Issues

The paper would be strengthened by insertion of numbers throughout to support ecological definitions and zonation. E.g. line 24, “substantial” – how much is substantial? A CO₂ degassing estimate from the literature would aid in contextualization. Or Ln 73 – what is “low temperature” in this context? What concentration of CO₂ is “heavy impact”?

“Interestingly” should not be used.

Ln 60 – list the specific pathways.

Ln 61 – this more hypothesized rather than certain.

Ln 74 – mofettes could be defined.

Ln 79 – briefly list the method used to show it was incorporated into biomass.

Ln 80-81 – “while the microbial ... controls.” Unclear what this means to have a community diversity be impacted by gases. Impacted how?

Ln 87 – “cold-water” list temperature.

Ln 89 – what is the tubing system? See major items.

Figure 1 – see carbon pathway comments. Also some colors are too dark and the font is too small to read at print size. Color may be better used for the different pathways than the different taxa since the discussion is more about that pathways. If nothing else, the bold lines demarcating the different metabolic pathways should extend the fill length of the figure.

Ln 143/44 – what does presence mean? Recruited reads from a given metagenome mapped to that genome? Should be clear in the caption how it was derived.

Ln 145 – brown and pink coloring not defined in the caption.

Ln 163/165 –conjecture should be moved to the discussion.

Figure 2 – what genes are used and what portion of the pathway was recovered?

Lns 318/319 – this would be highly dependent on the nutrient.

Ln 337 – what is the casing made out of?

Supplement:

Ln 114 – “at by”

Supplemental Table 3: Includes genomes below 70% completion and some that are 1.88 and 100. Should use the same metrics for completion from other studies and make sure that they are annotated correctly.

Reviewer #3 (Remarks to the Author):

To assess the impact of mantle degassing on subsurface microbial life under non-thermal conditions, this study used “...genome-resolved metagenomics to investigate how the gases impact the metabolism and activity of indigenous microbes compared to non-impacted sites. “

The primary results are as follows –

1. “...species-specific analyses of Altiarchaeota suggest site-specific adaptations and a particular biogeographic pattern, ...”

2. “...chemolithoautotrophic core features of the communities appeared to be conserved across 17 groundwater ecosystems between 5 and 3200 m depth.”

3. “We identified a significant negative correlation between ecosystem depth and bacterial replication, except for samples impacted by high amounts of subsurface gases, which exhibited near-surface activity.”

The conclusion drawn from Result 3 is that mantle degassing leads to higher microbial activity in the deep subsurface than previously estimated.

These are important results. Results 1 and 2 provide a nice large-scale overview of biogeographic variation and metabolic constancy in the continental subsurface. Result 3 is a novel and interesting advance in understanding of continental subsurface life.

I think an appropriately revised version of this manuscript would be very appropriate for publication in Nature Communications.

My primary concerns about the present manuscript are focused on the ‘stability’ of the Andernach ecosystem, the extent to which the sampled community represents the ecosystem, and the “in situ replication” results and discussion.

These concerns must be met before the manuscript is suitable for publication in Nature Communications. I think the manuscript can be revised fairly easily to meet them.

Comments –

Regarding stability and composition of the Andernach ecosystem:

The Andernach ecosystem is said to be “stable”. This statement needs to be more precisely phrased. The 15 years of data indicate that it is chemically and physically stable. However, it’s not clear that the system is biologically “stable” over time. Previous series have shown that community taxonomic composition in other subsurface aquifers changes over time. Because the Andernach samples are from two eruptions sampled on a single day (February 21, 2018), they do not show that the ecosystem is taxonomically stable over time.

Regarding the extent to which the sampled community represents the Andernach ecosystem:

Because the communities sampled at Geyser Andernach are planktonic, the extent to which they represent the entire in situ community (including biofilms) is not known. This point should be acknowledged.

The “reconstructed genome” is said to be “representative for the ecosystem”. This statement needs to be more carefully phrased, because the taxonomic composition of the ecosystem may change over time (above) and the samples are limited to planktonic organisms. It can be finessed by saying that the reconstructed genome is representative of the planktonic ecosystem at the time of sampling.

Regarding in situ replication:

“In situ replication indices” are central to the manuscript. However, they’re not explained anywhere in the manuscript. It appears to me that the manuscript is using the “Index of Replication (iRep) metric introduced by Brown et al. (2016). As described by Brown et al., iRep “...determines replication rates based on measuring the rate of the decrease in average sequence coverage from the origin to the terminus of replication. ... the iRep algorithm is distinct in that it makes use of the total change in coverage across all genome fragments”.

Based on my understanding of Brown et al. (2016), the discussion of replication in the present manuscript needs to be edited to account for the following points:

1. The technique must be clearly described before the data are introduced or interpreted. This description could be as brief as the lines from Brown that I quote above.
2. The result is a single index (not indices) and should be identified as such.
3. Despite the lines from Brown quoted above, the index measures a ratio, not a rate. In short, it provides an estimate of the proportion of cells that is replicating DNA. It does not provide an estimate of how long that replication takes. In principal, it seems to me that this replication could take years (or even never reach completion). This point should be briefly discussed.

Specific examples of this issue can be found in the lines referenced below.

Lines 127-128 – “in situ replication indices” are introduced without definition or explanation and interpreted to demonstrate that 40-50% of the population was dividing. Because this is a fundamental claim of the manuscript, what’s really being measured should be made clear before the claim of reproduction is made.

In lines 190-219, “in situ replication indices” are referenced as iRep and plotted in figure 3 but still undefined, and this claim of subsurface replication is extended to sites in different regions of Earth.

This problem is amplified by the fact that iRep is not even defined in the Methods, but simply referenced there to another publication that does not address indices of replication (Guindon et al., 2010)! The correct reference should be 40 (Brown et al., 2016).

Lines 282-284 – Sedimentary cells are not necessarily “sessile” (attached) and comparison to a few shallow sediment samples (Figure 4) does not necessarily “validate that replication indices can be used as proxies of activity in these [active aquifer] ecosystems.”

REVIEWER COMMENTS

Referee questions in black

Response by the authors in blue

Quoted new text in the manuscript in red

Reviewer #1 (Remarks to the Author):

This study presents metagenomic analysis of three groundwater samples from a CO₂-rich site, to explore if mantle-derived carbon stimulates a subsurface biosphere. This particular environment is enriched in Altiarchaeota, a candidate DPANN phyla without a representative isolate (*Candidatus Altiarchaeum hamiconexum* notwithstanding). The results of this study are compared to similar prior studies from other groundwater systems to examine if there are trends in deep biosphere diversity, growth, and metabolic function as a function of depth, geography, and/or environment type. Considering that continental microbes may comprise 60% of microbial biomass on Earth, this study could be of interest to a wide scientific community.

In my assessment, there are 4 major claims of this study, although they are not necessarily presented with equal weight in the presented manuscript: 1) That bacterial replication (and, by extension, activity) declines with depth, except in areas of mantle-derived C stimulation (mainly in Figure 3); 2) That there is little variation in metabolic potential despite geographic distribution patterns of taxa (Figures 2 and 4); 3) that *Altiarchaeum* from the new study site indicate an adaptation to hydrogen from the mantle by possessing an NiFe hydrogenase (Figure 2); and 4) that Altiarchaeota have a slow evolutionary rate of genetic divergence.

Response 1: We thank the reviewer for the supportive comments.

Regarding claim 1, I have concerns about the application of the iRep proxy for supporting this claim. The index of DNA replication (iRep) is a proxy for the instantaneous DNA replication rate of cells, calculated from the ratio of metagenomic read coverage near the origin of replication versus the terminus in individual genomes-from-metagenomes (i.e. MAGs), with assumptions made about the ordering of genomic bins based on coverage. While this approach has been validated with laboratory cultures, there is limited verification in environmental samples, and at least some indication that it is not a robust indicator of replication/growth (for example, Long et al. 2019 Biorxiv DOI: 10.1101/786939). Application of iRep to environmental samples may be complicated by factors like horizontal gene transfer, or close relatives cross-recruiting mapped reads, or extracellular DNA, that impact read recruitment but are not indicators of population ecology. Its application may also be complicated by a lack of understanding of the relationship of replication and growth in slowly growing populations replicate under energy-limiting conditions, as would be the case in the subsurface. Furthermore, the assumption that all cells in each sample are in physiological states representative of in situ conditions will be dependent on sampling strategies, which is not controlled in the meta-analysis (at least, not indicated in Supplemental Table 6). Also problematic to me is that the study only calculated iRep values for 10 of the 52 microbial groups claimed to be in the community (Figure 1); how then can it be asserted that 40-50% of the population was undergoing replication when there are not values for 80% of the population? Thus, while there are clear differences in the iRep values of different samples, and that the samples exposed to mantle-derived carbon have higher values (i.e. Fig 3), I am not convinced that the interpretation of those is as robust as the authors claim (i.e. that it indicates that “geological degassing leads to...higher microbial activity” L34; L285:

"This finding validates that replication indices can be used as proxies of activity in these subsurface ecosystems" L 286: "Using these measures across multiple samples from different depths established that the activity of bacteria declines with sampling depth, suggesting a deep biosphere with little activity"). Because there are possible confounding factors that could influence the iRep value, as indicated above, I think it is speculative to state that these values are indicators of activity. I think that the authors need to more cautiously interpret the variation in iRep values to account for these possible confounding factors, and to considerably tone down the use of "activity" as a replacement for "replication" (since some reviewers, like myself, equate activity with a time-dependent rate).

Response 2: We have addressed all concerns of the reviewer in the revised manuscript and would like to list the individual aspects here as bullet points:

- We agree that iRep is a measure of genome replication rather than microbial activity. We have made corresponding changes throughout the manuscript. In detail, we rephrased iRep to be a measure of genome replication (as opposed to dormancy) instead of activity and added a supplemental discussion chapter discussing the use of iRep to assess active replication (see suppl. Discussion L186-218).
- Regarding the confounding factors (HGT, external DNA etc.) that might impact iRep, we have emphasized in the manuscript the fact, that we do not use iRep as an individual measure of bacterial replication rather as a measure across 895 bacterial species for performing statistics. Given this high number of replication measures that we leveraged individual aspects like HGT affecting one species in the dataset is in our opinion neglectable. In the particular case of HGT, we would like to note that if these are in large genome fragments surrounded by host genes, it would not affect the measured iRep value. If the genes affected by HGT are on smaller fragments, they would not be considered in our binning approach since all genomes undergo GC and coverage correction (see Bornemann et al., 2020¹).
- Regarding the question of the 40-50% measured iRep values and the conclusion that 80% of the population is replicating: This was a miscommunication, and we have corrected this statement as follows: "We verified that bacteria in this community are replicating using *in situ* replication index values. Replication index values are calculated from the difference of sequencing coverage between the origin of replication and terminus of replication. Proliferating organisms replicate their genomes with multiple replication forks starting at the replication origin and thus contributing more to sequencing reads. In our study, these index values ranged between 1.4-1.5, indicating that 40-50 % of those microbial populations, whose iRep values were calculated, underwent genome replication at the time of sampling." (L129-135).
- In addition, we followed the suggestion by Long et al. (2020)² and used the measure of codon usage bias to determine the maximum possible growth rate. However, we would like to note that this "growthpred" value is static for a single species irrespective of its activity. We also determined a linear relationship between sampling depth and the respective values. However, the analysis determined that the deeper the sampling site the greater the maximum growth rate, i.e. the response of the community is faster. These results align well with the fact that they are overall slower growing as nutrients are low and microbes need to respond to nutrients faster in these ecosystems (see L177-183 for results, Fig. S4 for a graphical comparison of iRep vs growthpred and L449-455 for the methodology on calculating maximal growth rates). This underpins a recent study (Mehrshad et al. (2020)³), in which the authors postulate a "halt and catch fire" scenario for microorganisms in deep aquifers. When transferring the analysis of iRep and growthpred to sites impacted by geological

degassing – which have in theory higher nutrient input – we also determined higher iRep values and higher growthpred values underlining our overall statement that sites impacted by geological degassing are similar to near surface ecosystems.

Beyond these concerns about whether or not the iRep are actually indicating changes in replication with depth, I encourage the authors to put their discussion about changes in the deep biosphere with depth in the perspective of other recent papers on this topic, such as D’Hondt et al. 2019 Nature Communications, Kirkpatrick et al. 2019 Frontiers, Lloyd et al. 2020 Biorxiv, Starnawski et al. 2017 PNAS.

Response 3: We have taken these studies into account as follows: We incorporated the findings of Kirkpatrick et al. (2019)⁴, Lloyd et al. (2020)⁵ and Starnawski et al. (2017)⁶ into the discussion section. We would like note, that we initially didn’t cite these studies as they focus on the marine deep biosphere or subduction zones. The newly added lines state (L259-270):

“Prior studies^{4,6} observed a reduction in microbial load with marine sediment depth and age, indicating that communities in older sediments were probably formed by members of surface communities that have a higher degree of persistence compared to others. Thus, subsurface communities would not be formed by actively replicating organisms but instead be shaped by the differing mortality of surface community members^{4,6}. The upper ten centimeters of sediment were found to be an exception showing active proliferation⁷. Although we analyzed many different ecosystems, our data do not allow drawing conclusions about the impact of mortality shaping subsurface microbial communities as they originate from different geologic formations. However, our observed decrease in replication measures with sampling depth does agree with these prior observations of a reduction of microbial load with depth and indicate that replication is occurring, albeit with less replication forks in the subsurface compared to near-surface ecosystems.”

We decided against adding the findings of D’Hondt et al. (2019)⁸ as we do not have identical types of geochemical data for the various ecosystems to be able to compare their geochemistry and we do not want to make any generalized statements about the impact of those microbes on global biogeochemical cycles.

Regarding the 2nd major claim, that there is little variation in metabolic potential despite geographic distribution patterns of taxa (Figures 2 and 4), this is not a surprising outcome to me considering other studies that have documented metabolic redundancy even when diversity fluctuates, including in the subsurface such as Tully et al. 2017 The ISME Journal.

Response 4: We have incorporated the suggested literature as a comparison into L196-199, stating:

“Consequently, and in congruence with previous studies investigating the metabolic diversity in a seafloor aquifer⁹, little difference exists in the metabolic potential between regular subsurface microbial communities and those at sites impacted by mantle degassing, although the indigenous organisms at these sites appear to have higher replication index values.”

Regarding the 3rd claim, that Altiarchaeum from the new study site indicate an adaption to hydrogen from the mantle by possessing an NiFe hydrogenase (Figure 2), these species-specific differences are mainly addressed in the results. If the authors tone down the iRep-based inferences in the paper, this could open up space for digging deeper into Altiarchaeota-specific biogeography and functional variation within the manuscript.

Claim 4, that Altiarchaeota have a slow evolutionary rate of genetic divergence (for example, L187), isn't brought up in the abstract, but it is sizeable point of discussion in the paper and supplement. I find this discussion interesting, and encourage the authors to pursue it even more. As above, if a revised paper de-emphasizes the iRep-based claims, then this could open up space for presenting a deeper presentation of the Altiarchaeota.

Overall, I found this paper to compress two stories together – one about replication/activity in the subsurface, and the other about Altiarchaeota biogeography. As expressed above, I have reservations about the assumptions and interpretations of the replication-oriented story. By contrast, I think the Altiarchaeota focus story could be further developed. Therefore, I recommend that the authors de-emphasize the replication-based story, and be more conservative in the interpretations based on the caveats of the assumptions of what iRep is actually measuring, and instead shift the focus of the paper to the Altiarchaeota.

Response 5: We have reduced the amount of discussion directed towards the iRep findings. Instead, we now have conducted comparative genomics of Altiarchaea, which resulted in the finding that their genome fluidity is mainly impacted by gene loss or horizontal gene transfer, possibly counteracting the slow evolutionary rate. Please see the new Figure 5 and the expansion upon results chapter (L201-246) “Biogeography and functional adaptations of deep subsurface Altiarchaeota” as well as the new discussion sections discussing Altiarchaeota genome fluidity and biogeography, stating (L301-331):

“The abovementioned hypothesis regarding replication speed of *Ca. Altiarchaea* would also align well with their strict biogeography. The clustering by continent of origin (North America, Europe, Asia), also reproducible in ANI and AAI (Fig. S7), indicate a strict provincialism. As dispersal via the surface is unlikely due to the high oxygen sensitivity of *Ca. Altiarchaea*¹⁰, plate tectonics could have been a viable alternative dispersal route providing ample opportunities for the common ancestor to distribute to North America and Europe. Plate tectonics have recently been implicated as the potential dispersal route for *Ca. Desulforudis audaxviator* to Africa, North America and Eurasia between 55 to 165 Myr¹¹. The dispersal of *Ca. Altiarchaea* could have occurred within the Phanerozoic, starting with the early Devonian (~400 Myr), when the continental margins Laurentia and Baltica, which form today's North America and Europe, respectively, collided to form Laurasia^{12,13}. Japan, on the other hand, has not been in contact with those margins since the break-up of Rodinia 750-600 Myr ago¹⁴, thus making dispersal to Japan during the Phanerozoic unlikely. As European and Japanese *Ca. Altiarchaea* are indicated to have a common ancestor, one possible route of dispersal from Europe to Japan could be across the Siberian plate through China in the early Mesozoic and then transferal to Japan during the plate processes, which uplifted the Japanese islands from the sea 25 Myr ago. Future studies are necessary to recover *Ca. Altiarchaea* genomes from Asia to further underpin this hypothesis of dispersal, since current public datasets from this continent are substantially underrepresented in databases.

The strict biogeography of the *Ca. Altiarchaea* is reflected by the conserved core metabolism, with most pathways being present in every *Ca. Altiarchaea* genome and indicate a slow evolving genus. However, observed putative gene loss and gene transfer events in investigated *Ca. Altiarchaea* populations indicate a compensatory strategy to counteract the slow evolutionary rate. This observed gene loss and transfer might be exuberated by the exclusive living in biofilms, which have generally been known as hotspots of HGT for Bacteria¹⁵. The genes in *Ca. Altiarchaea* acquired via HGT are mainly from the

bacterial domain, an evolutionary process frequently occurring in nature¹⁶. This HGT likely took place in the subsurface due to the immobility of *Ca*. Altiarchaea mediated by the anchoring of cells via their *hami*. Consequently, our analyses provide evidence that subsurface ecosystems impacted by geological degassing can be hotspots of microbial life and of increased evolutionary rates bolstered by lateral gene transfer across domains.”

Despite my concerns about the interpretations, I do commend the authors on writing a very nice manuscript; it was easy and engaging to read. I caution the authors to carefully review the citation callouts, as there seems to be some mismatches between the numbered references and the actual citations I think they meant to callout.

Response 6: We have corrected these mistakes.

Overall the methodological details provided were appropriate (except where noted below), although I must flag that the authors did not provide the SRA accession numbers for their metagenomic sequence data (only indicated as "XXX" in Supplemental Table S5, and not mentioned in the text), so it was impossible for the reviewer to evaluate the data unless knowing where to find the data within the Census of Deep Life. This makes reviewing a sequence-based manuscript very difficult, if not impossible, to verify the data processing steps. As a reviewer, I cannot recommend accepting a manuscript that does not provide access to all of the data. Also, it does not appear that the authors included a no-template-control in their sequencing, which is problematic for a low biomass study (Sheik et al. 2018 Frontiers).

Response 7: We have replaced the “xxx” in the Supplementary Table S5 with the respective accessions. We performed no non-template control as negative control samples did not result in measurable DNA, and we were consequently neither able to reconstruct a library nor sequence it.

Minor suggestions:

L19: consider adding additional references to support the statement that "releases huge amounts of oxidized carbon...into the crust and atmosphere", such as Werner et al. 2019 (<https://www.cambridge.org/core/books/deep-carbon/carbon-dioxide-emissions-from-subaerial-volcanic-regions/F8B4EFAE0DAF5306A8D397C23BF3F0D7>), to support the one field-based study cited.

Response 8: We have added the supplied literature as an additional reference, stating (L60-62):

“One source of such gases can be Earth’s mantle, which also releases 38.7 ± 2.9 Tg/yr of oxidized carbon¹⁷, mainly in form of carbon dioxide (CO₂), into the crust and the atmosphere^{18,19}.”

L72: Consider also adding the Fullerton et al. 2019 study from Costa Rica mantle degassing systems, as another example: <https://eartharxiv.org/gyr7n/>

Response 9: We have added the reference as an added example in L65-66, now stating: “Modern Earth has few areas with active mantle degassing, which are usually restricted to terrestrial volcanoes, subduction zones or hydrothermal vents in oceans^{20,21,22,23,24}.”

L119: The manuscript states that 15 organisms were detected in all three metagenomes, but

when I look at Figure 1, I only count 7 rows with genomes in each three samples. What am I missing?

Response 10: The 15 common organisms mentioned here were detected via *rpS3*. Not for all of those, genomes could be reconstructed, particularly in all samples. We've rephrased the sentence to make it clear that the 15 common organisms were detected via *rpS3*. The rephrased sentence states (L120-122):

“The core community was composed of 15 organisms detected via *rpS3* across all three metagenomes (Fig. 1), and they accounted for 42.8% (1.3% SD) of the total relative abundance of the community.”

L122 (and L240 and L269 and L308): "Interestingly" is subjective (see <https://www.nature.com/articles/ni1105-1061> for more discussion of not using such “helper words” in scientific writing); I suggest removing this word.

Response 11: We have removed the filler word from the named lines. We also removed the word “Interestingly” from additional lines L203 and L292.

L128: Based on Figure 1, you only calculated iRep values for 10 of the 52 microbial groups you claim are in the community. Since you do not have these estimates of this index for ~80% of the community members, I think it is erroneous to say that 40-50% of the population was undergoing replication at the time of sampling. That metric can only be applied to the 20% of taxa that the estimate could be made for. Please rephrase this statement accordingly.

Response 12: As indicated above, this was a miscommunication and we have rephrased the section accordingly (L133-135):

“In our study, these index values ranged between 1.4-1.5, indicating that 40-50 % of those microbial populations, whose iRep values were calculated, underwent genome replication at the time of sampling”

L135: "t" tonne is not an SI unit. I am assuming you mean a metric ton, which is 1000 kg. Please use proper SI units through the manuscript and supplemental text. For example, 6.27×10^3 kg. This will also make the comparison to the calculation of microbial biomass easier for the reader.

Response 13: We have changed t to kg, C to K and hectare to m^2 . We have decided against changing time units (minutes, hours, days, months and years) to seconds as well as liters to dm^3 as we feel that these changes would impact the readability negatively. (L71, L73, L86, L141).

L282: I don't think you have the correct reference cited for the statement about higher replication rates in sediment versus plankton? Hoang et al. 2018?

Response 14: The citation callouts seem to be off in the initial submission. We apologize for the confusion and have corrected this here (L149-151).

L284: I don't think that Figure S4 is the correct callout to support this statement? Do you mean Figure 3?

Response 15: Yes, we indeed meant Figure 3 (Figure 2 in the revised manuscript).

L300: The argument about plate tectonic driven dispersal of Alti-1 is highly speculative, so I don't think that "most plausible mechanism" is the correct phrasing here. I suggest "a possible mechanism".

Response 16: We weakened the interpretation of the dispersal of Altiarchaea and discussed it now in more detail to avoid turning speculation into facts (L304-319):

“As dispersal via the surface is unlikely due to the high oxygen sensitivity of *Ca. Altiarchaea*¹⁰, plate tectonics could have been a viable alternative dispersal route providing ample opportunities for the common ancestor to distribute to North America and Europe. Plate tectonics have recently been implicated as the potential dispersal route for *Ca. Desulforudis audaxviator* to Africa, North America and Eurasia between 55 to 165 Myr¹¹. The dispersal of *Ca. Altiarchaea* could have occurred within the Phanerozoic, starting with the early Devonian (~400 Myr), when the continental margins Laurentia and Baltica, which form today’s North America and Europe, respectively, collided to form Laurasia^{12,13}. Japan, on the other hand, has not been in contact with those margins since the break-up of Rodinia 750-600 Myr ago¹⁴, thus making dispersal to Japan during the Phanerozoic unlikely. As European and Japanese *Ca. Altiarchaea* are indicated to have a common ancestor, one possible route of dispersal from Europe to Japan could be across the Siberian plate through China in the early Mesozoic and then transferal to Japan during the plate processes, which uplifted the Japanese islands from the sea 25 Myr ago. Future studies are necessary to recover *Ca. Altiarchaea* genomes from Asia to further underpin this hypothesis of dispersal, since current public datasets from this continent are substantially underrepresented in databases.”

L324: this sentence about Early Earth is HIGHLY speculative. There are so many factors to consider that would influence microbial activity, besides just mantle degassing driven stimulation, that it is a wild guess if there was a "high microbial activity" at the surface. I think that this speculation detracts from the findings and strongly suggest removing it.

Response 17: We have removed the predictions regarding Early Earth as suggested.

L376: Was a no-template control sequenced in this study? Some of the proteobacterial clades contain taxa that are commonly detected as lab contaminants in sequencing efforts that have proper controls (*Pseudomonas*, *Sphingomonas*). The authors should compute ANI values for these bins compared to genomes of cultures / MAGs / SAGs in NCBI/IMG via a tool like FastANI to understand and communicate how likely these bins are contaminants. Any Bin with a suspiciously high ANI value to anything that is not subsurface should be removed and all downstream statistics would need to be recalculated.

Response 18: As mentioned in Response 7, the non-template controls exhibited no measurable amounts of DNA and we consequently were not able to sequence them. In order to detect potential contaminant genomes recovered from Geyser Andernach, we calculated the ANI of genomes recovered from GA to the GTDB reference genome database using the GTDB-tk classify_wf workflow. We assumed that variants of contaminant genomes would already be included in the GTDB and thus potential contaminant genomes of Geyser Andernach would display a high ANI value to a reference genome. The only genome with an ANI greater than 80 % (lower ANI values are not reliable as distance metrics and thus not reported by GTDB-tk) was

GA_180221_E-1-2_metaspades_Carnobacterium_36_4. This genome was consequently excluded from the analyses as it is assumed that contaminant species are deposited into databases. Details of this analysis were incorporated into the methods section L426-433, stating:

“**Identification of potential contaminant genomes.** The GTDB-Tk²⁵ classify_wf workflow with default parameters was used to place the recovered genomes from the Geyser Andernach in relation to a reference dataset. If a close relative genome was identified in this approach, we calculated the ANI between the reference and the newly recovered genome. The only genome showing a similarity ≥ 80 % ANI to the reference dataset was GA_180221_E-1-

2_metaspades_Carnobacterium_36_4 (96.42 % ANI to *Carnobacterium alterfunditum* GCF_000744115.1) and was thus identified as a potential contaminant and excluded from further analyses.”

Additionally, a second sampling was performed in March 2019, one year after the original sampling. For this sampling, a valve connecting to the geyser was used to tap into the geyser directly. This made sure that possible surface contaminations, e.g., erupted water flowing back into the geyser were not a factor for this sampling. Mapping the reads of the metagenome from March 2019 back to the dereplicated genomes from the original sampling showed that 73.82 % of the reads were recruited by those genomes (see Figure S12). This makes us confident that these genomes do belong to organisms from the subsurface and do not constitute surface contaminants. Please see response 46 for a more detailed comparison of the metagenomes from 2018 and 2019 as well as further discussion regarding the stability of the ecosystem.

L381: It is unclear whether the samples from this current study were “co-assembled” (reads from DNA extracts from all three filters assembled together) or assembled one at a time. Line 58 in supplemental methods uses the phrase “differential coverage binning” which suggests they were co-assembled but the wording isn’t clear prior to that.

Response 19: We did not co-assemble the samples and rephrased the methods to reflect that (L394), stating:

“The three samples were processed individually as follows.”

With differential coverage binning we refer to the cross-mapping of reads of the different samples on each separately assembled assembly of the same ecosystem, thus resulting in three coverage profiles for each scaffold in each sample and using this for the respective binning tool.

Figure 1: the text is impossible to read in this two-column figure. Consider making the phylogenetic tree branching portion of the image thinner in horizontal space to allow more room for text. Also, the figure does not have A and B panel labels, although the figure caption indicates an A and B. Furthermore, as presented, lining up the DAPI images vertically suggests that the morphologies are somehow connected to specific phyla, which is not the intention that I think you mean to convey. Consider moving the DAPI images to be in a row below the metagenomic figure panel (which again will provide more space to make the text larger).

Response 20: We have modified Figure 1 accordingly.

Figure S4: This is a weak correlation with a high p-value and it looks like it is really influenced by the higher number of near-surface samples and much lower amount of deeper samples.

Response 21: We are aware that the correlation is not highly significant and that there are more near-surface samples than deep subsurface samples, potentially making the diversity estimations for deeper samples likely less accurate as less data points are used for the correlation. However, since this is not driving the main story of the manuscript we are satisfied with a p-value of 0.02 particularly given the fact that the deep subsurface is so heavily underexplored as documented herein. Additionally, we would like to note that a p-value < 0.02 is generally telling a significance and the chance that the result is wrong is below 2%.

Reviewer #2 (Remarks to the Author):

Summary

Bornemann et al. performed metagenomic analysis, along with collecting ancillary geochemistry, on a geyser system. It provides novel information on a cooler endmember of such systems and is contextualized with results from other subsurface studies to interpret the role of carbon fixation utilizing the high CO₂ in these waters. The manuscript contains a lot of work and a lot of novel metagenomic data. However, the current presentation overstates claims and does not provide enough information to allow the reader to evaluate interpretations, as detailed below.

Response 22: We thank the reviewer for the critical assessment of our study.

Major Issues

Pathways and genes used: Since it is a major component of the paper, the specific genes used to determine metabolic capabilities should be included in the manuscript rather than just a reference to another paper. The figures should list the specific gene used and there should be a table of which genes were used from each metagenome to determine all of the metabolisms discussed in the paper. This would also add clarity for if only one of the genes was used to represent a given metabolism, or if all were required to declare presence of the metabolism. Were any complete pathways recovered? There are also nuances to the genes used in Anantharaman et al 2016. For example, from your reference's SD14, propionyl-CoA synthase used for 3HP cycle is a gene that is also used in other pathways than 3HP, which is another reason to be explicit in this work which genes were used.

Response 23: As suggested, we added supplementary Table S10 listing the specific HMMs as well as their pathways and have elaborated in the supplementary methods on how pathway abundances were determined, stating (L68-79):

“Metabolic potential predictions. Hidden Markov-models (HMMs) and their corresponding score thresholds originating from Anantharaman et al. (2016)²⁶ were used to identify key enzymes in chemolithoautotrophic pathways (see table S10 for a list of HMMs, their respective score thresholds and the pathways they are indicative of). If multiple genes indicative of a single pathway were recovered in assemblies, their abundances were aggregated by determining the maximum abundance. For the prediction of the metabolic potential for sulfur oxidation / reduction on genomes, many of the included pathways require specific genes to be absent while others need to be present to indicate the presence of the specific pathways. Thus, we used the following presence/absence scheme for the HMMs detailed in table S10 to identify present pathways in genomes.

Sulfide oxidation	sulfide_quinone_oxidoreductase_sqr OR fccB need to be present
Sulfite reduction	dsrA AND dsrB AND dsrD need to be present
Sulfur oxidation with dsr	dsrA AND dsrB need to be present but dsrD needs to be absent
Sulfur oxidation with sor	sor needs to be present
Sulfur oxidation with sdo	sdo needs to be present
Sulfate reduction with asr	asrA AND asrB AND asrC need to be present
Sulfate reduction with aprA and sat	aprA AND sat need to be present

We also added the gene names to the captions of Figure 1 and Figure 2 (Figure 4 in the revised manuscript) to be more explicit in which gene is indicative of which pathway. Additionally, Supplementary Figures S5 and S6 were added, displaying the individual gene hits and their coverages used to construct Figure 4 (Figure 3 in the revised manuscript).

Activity: The interpretation of activity is overstated since only a proxy for activity was used. iRep metrics could be used in conjunction with activity findings by other means (e.g. SIP or metaT), but cannot be used on their own to support the claims of microbial activity in the abstract and throughout. This is especially true given its use on uncomplete genomes, where the metric was only 70% completion or above, and the 5m sediment vs water comparison has enough variability that it does not suffice in place of an actual activity measurement. Therefore, the manuscript should be revised to describe findings as a potential proxy for activity not an actual measurement of activity.

Also as no activity measurements were conducted, metagenomic analysis should be more clearly couched as potential carbon fixation/carbon fixation potential rather than definitive active carbon fixation.

Response 24: We rephrased the manuscript and explain that iRep is a **proxy** for activity and not a direct measure of activity. We would also like to point to Response 2 for a list of changes to the iRep component of the manuscript, which we drastically weakened.

Regarding the metabolic potential for carbon fixation predicted from metagenomes, we have now rephrased the statements to be more explicit that we are talking about a potential rather than an actual measurement of activity.

Carbon in cells: 90 fg C/cell is high for a subsurface ecosystem. See Kallmeyer et al. 2012 PNAS for a more recent discussion of the choices involved in choosing a number for this in subsurface systems.

Response 25: We have adjusted the estimate of intracellular carbon per cell to 14 fg /cell to more appropriately reflect the carbon content of microbial cells in the subsurface according to Kallmeyer et al. (2012)²⁷.

System plumbing and contamination controls: Cell densities are very high for subsurface fluids. While it appears the recovered organisms that are the focus of the paper are environmentally relevant and not common contaminants, Figure 1 contains some groups that are often contaminants (Salter et al). It is important to constrain where in the system the Alt. archaea thought to be located to support the interpretation of their role in carbon cycling. Are there areas where stagnant waters could collect between sampling? Are they flushed for a period of time with some metric to ensure formation waters are being sampled? Are biofilms able to grow along sampling lines and these cell densities are instead measurements of biofilms? A diagram of the mechanics of the sampling system/plumbing with a depiction of where microbial communities are expected to reside within this system and a discussion of how sampling of formation fluids has been verified are required for interpretation of these results.

Response 26: For the contamination questions, we would like to refer here to response 18, where we addressed these concerns in detail.

Regarding the type of water sampled: The upper 83 m of the geyser well have a casing and are sealed with cement so that no water can enter the system from the sides. The residual length of the geyser borehole (83-351.5 m) is intermittently covered by bridge-slotted screens which allow entry of CO₂-saturated water (along with containing microorganisms) into the

geyser well. We sampled two sequential eruptions, with about 6000-7000 L of water being flushed through the tubing of the geyser per eruption. The geyser tubing (cylindric shape, diameter 15 cm and 351.5 m deep) holds approximately 6212 L of water. Consequently, each eruption flushes the tubing system once. Prior to the sampled eruptions, a test eruption was performed, thus flushing the tubing of any residual water. This ensured that the sampled community really stems from the deep subsurface. This also includes the biofilms of the Altiarchaeota, which substantially contributed to the cell counts of course. We also added another sampling event to the revisions, for which we collected samples directly from the geyser outlet after flushing. Despite this, the communities between the eruptions remain identical, indicating that the communities sampled during both eruptions are formation waters (Fig. S12). This information was added to the methods section (L380-388) and states:

“The upper 83 m of the geyser well have a casing and are sealed with cement so that no water can enter the well from the sides. The residual length of the geyser borehole (83-351.5 m) is intermittently covered by bridge-slotted screens which allow entry of CO₂-saturated water into the geyser well (Fig. S1). Each eruption flushes the tubing system (cylindric shape, 7.5 cm radius, 351.5 m length, approximate volume 6.2 m³) with 6-7 m³ water and an additional eruption was performed prior to the sampled eruptions to rid the tubing system of any stagnant water. The metagenomes recovered from both eruptions show identical community compositions and consequently, the sampled communities should be representative of subsurface communities and not contamination from the tubing system.”

We have also incorporated a diagram of the subsurface environment of the Geyser Andernach as Figure S1. We do expect that microbes can grow on all surfaces within the ecosystem (faults and quartz veins as well as in the sediment). Thus, we’ve decided against incorporating them into the diagram.

Minor Issues

The paper would be strengthened by insertion of numbers throughout to support ecological definitions and zonation. E.g. line 24, “substantial” – how much is substantial? A CO₂ degassing estimate from the literature would aid in contextualization. Or Ln 73 – what is “low temperature” in this context? What concentration of CO₂ is “heavy impact”?

Response 27: We have added concrete values as suggested. Please see lines L25, L61, L71 and L86, stating the following respectively:

“Earth’s mantle releases 38.7 ± 2.9 Tg/yr CO₂¹⁷ along with other reduced and oxidized gases ...”

“One source of such gases can be Earth’s mantle, which also releases 38.7 ± 2.9 Tg/yr of oxidized carbon¹⁷, mainly in form of carbon dioxide (CO₂), into the crust and the atmosphere^{18,19}.”

“... little is known about deep subsurface ecosystems with low temperatures (283-293 K) and still impacted by gases released from the mantle ...”

“The cold-water (291 K) Geyser Andernach is located in the Rhine Valley near Koblenz in western Germany and is driven by gases discharged from the mantle¹⁸.”

Note that we refrained from concretizing “heavy impact” and instead just state “... and still impacted by gases released from the mantle...” as we did not find a suitable definition of what a heavy impact in this context is.

“Interestingly” should not be used.

Response 28: We have removed all occurrences of ‘interestingly’ accordingly.

Ln 60 – list the specific pathways.

Response 29: We have added the specific pathway, now stating (L56-57):
“*Ca. Altiarchaeota live autotrophically using the Wood-Ljungdahl carbon fixation pathway*²⁸,...”

Ln 61 – this more hypothesized rather than certain.

Response 30: We have removed the hypothesis that the WL-pathway is the most dominant carbon fixation pathway prior to the evolution of photosynthesis.

Ln 74 – mofettes could be defined.

Response 31: We have added a definition of mofettes in L72-73, stating:

“Previous studies have analyzed the influence of mantle degassing via volcanic mofettes, i.e. carbon dioxide seeps below 373 K,...”

Ln 79 – briefly list the method used to show it was incorporated into biomass.

Response 32: Done as suggested in L76-79:

“Beulig and co-workers reported an increase in dark carbon fixation and found evidence that the CO₂ from the degassing is indeed incorporated into biomass based on IR-GC/MS measurements of fatty acid methyl-esters and DNA Stable-Isotope Probing experiments of microcosms fed with ¹³C-labelled CO₂^{29,30}.”

Ln 80-81 – “while the microbial ... controls.” Unclear what this means to have a community diversity be impacted by gases. Impacted how?

Response 33: We have rephrased the sentence (see L79-81) to make clear that mofettes and reference soils were compared. The rephrased sentence now states:

“Along with fermentation processes, the pathways for the turnover of organic carbon were similar in both systems, while the microbial diversity of soils in mofettes was lower compared to controls.”

Ln 87 – “cold-water” list temperature.

Response 34: We have added the temperature information to the section. As “cold-water geyser” is a defined geyser category as opposed to the classical geysers where steam is the driving force behind the eruption, we have decided to only slightly modify the section to “The cold-water (291 K) Geyser Andernach ...” (L86)

Ln 89 – what is the tubing system? See major items.

Response 35: We have added a diagram of the plumbing/tubing system of the Geyser Andernach ecosystem as Figure S1.

Figure 1 – see carbon pathway comments. Also some colors are too dark and the font is too small to read at print size. Color may be better used for the different pathways than the different taxa since the discussion is more about that pathways. If nothing else, the bold lines demarcating the different metabolic pathways should extend the fill length of the figure.

Response 36: We have modified Figure 1 accordingly.

Ln 143/44 – what does presence mean? Recruited reads from a given metagenome mapped to that genome? Should be clear in the caption how it was derived.

Response 37: We have rephrased the respective passages to more clearly convey how we inferred the presence of genomes and pathways. The caption now states (L783-785):

“*Matching recovered draft genomes in each sample (A, B and C for samples GA_E1-1, GA_E1-2 and GA_E2-1 respectively), i.e. genomes binned from these samples, are provided*”

as green boxes (otherwise left white). The presence of marker genes based on a marker gene search using HMMs on these genomes for specific chemolithoautotrophic pathways is shown as green boxes (otherwise left white)."

Ln 145 – brown and pink coloring not defined in the caption.

Response 38: We have added the definition to the caption. We would like to note that we revised Figure 1 according to response 20 and also modified the colors in the heatmaps for genomes and genes used. Now, green shows presence of genome or gene, respectively, while the field being left white signifies their absence. Please see response 37 for the respective revised passage of the caption.

Ln 163/165 –conjecture should be moved to the discussion.

Response 39: We have moved the discussion regarding dispersal of *Alti-1* to the discussion (L301-319), stating:

*"The abovementioned hypothesis regarding replication speed of *Ca. Altiarchaea* would also align well with their strict biogeography. The clustering by continent of origin (North America, Europe, Asia), also reproducible in ANI and AAI (Fig. S7), indicate a strict provincialism. As dispersal via the surface is unlikely due to the high oxygen sensitivity of *Ca. Altiarchaea*¹⁰, plate tectonics could have been a viable alternative dispersal route providing ample opportunities for the common ancestor to distribute to North America and Europe. Plate tectonics have recently been implicated as the potential dispersal route for *Ca. Desulforudis audaxviator* to Africa, North America and Eurasia between 55 to 165 Myr¹¹. The dispersal of *Ca. Altiarchaea* could have occurred within the Phanerozoic, starting with the early Devonian (~400 Myr), when the continental margins Laurentia and Baltica, which form today's North America and Europe, respectively, collided to form Laurasia^{12,13}. Japan, on the other hand, has not been in contact with those margins since the break-up of Rodinia 750-600 Myr ago¹⁴, thus making dispersal to Japan during the Phanerozoic unlikely. As European and Japanese *Ca. Altiarchaea* are indicated to have a common ancestor, one possible route of dispersal from Europe to Japan could be across the Siberian plate through China in the early Mesozoic and then transferal to Japan during the plate processes, which uplifted the Japanese islands from the sea 25 Myr ago. Future studies are necessary to recover *Ca. Altiarchaea* genomes from Asia to further underpin this hypothesis of dispersal, since current public datasets from this continent are substantially underrepresented in databases."*

Figure 2 – what genes are used and what portion of the pathway was recovered?

Response 40: We would like to refer to response 23.

Lns 318/319 – this would be highly dependent on the nutrient.

Response 41: The respective section was rephrased due to other corrections we made (L279-283):

"In these fracture-controlled aquifers, which are characterized by solid rock formation-embedded channels, flows can reach up to multiple magnitudes greater speeds than flows in comparable sediment-hosted aquifers. Thus, the availability of reduced mantle gases like H₂ and H₂S as microbial electron donors highlight the absence of nutrient bursts and the presence of a continuous nutrient flow similar to biomes on Earth's surface."

Ln 337 – what is the casing made out of?

Response 42: Stainless steel. For more information about the tubing system, we refer to response 26 and Figure S1.

Supplement:

Ln 114 – “at by”

Response 43: Corrected.

Supplemental Table 3: Includes genomes below 70% completion and some that are 1.88 and 100. Should use the same metrics for completion from other studies and make sure that they are annotated correctly.

Response 44: We have made sure that they use the same format and are annotated correctly.

Reviewer #3 (Remarks to the Author):

To assess the impact of mantle degassing on subsurface microbial life under non-thermal conditions, this study used "...genome-resolved metagenomics to investigate how the gases impact the metabolism and activity of indigenous microbes compared to non-impacted sites."

The primary results are as follows –

1. "...species-specific analyses of Altiaarchaeota suggest site-specific adaptations and a particular biogeographic pattern, ..."
2. "...chemolithoautotrophic core features of the communities appeared to be conserved across 17 groundwater ecosystems between 5 and 3200 m depth."
3. "We identified a significant negative correlation between ecosystem depth and bacterial replication, except for samples impacted by high amounts of subsurface gases, which exhibited near-surface activity."

The conclusion drawn from Result 3 is that mantle degassing leads to higher microbial activity in the deep subsurface than previously estimated.

These are important results. Results 1 and 2 provide a nice large-scale overview of biogeographic variation and metabolic constancy in the continental subsurface. Result 3 is a novel and interesting advance in understanding of continental subsurface life.

I think an appropriately revised version of this manuscript would be very appropriate for publication in Nature Communications.

My primary concerns about the present manuscript are focused on the 'stability' of the Andernach ecosystem, the extent to which the sampled community represents the ecosystem, and the "in situ replication" results and discussion.

These concerns must be met before the manuscript is suitable for publication in Nature Communications. I think the manuscript can be revised fairly easily to meet them.

Response 45: We thank the reviewer for the supportive comments and on a potentially revised manuscript.

Comments –

Regarding stability and composition of the Andernach ecosystem:

The Andernach ecosystem is said to be "stable". This statement needs to be more precisely phrased. The 15 years of data indicate that it is chemically and physically stable. However, it's not clear that the system is biologically "stable" over time. Previous series have shown that community taxonomic composition in other subsurface aquifers changes over time. Because the Andernach samples are from two eruptions sampled on a single day (February 21, 2018), they do not show that the ecosystem is taxonomically stable over time.

Response 46: We have sequenced an additional sample taken in March 2019 (just over one year after the initial sampling) to further investigate the stability of the ecosystem. For this

sampling, we refined the sampling approach by utilizing an outlet connected to the geyser tubing. This ensures that there is little to no contamination from air throughout the eruption. This outlet allowed us to tap into the geyser without requiring us to wait for and collect erupting water as the gas continuously pushed out water through the valve and the connecting tube. This avoided any potential sources of contamination from the surface. We compared the newly sequenced sample with the sequences we received one year previously by both mapping the reads on the dereplicated set of genomes (see Fig. S12). In total, this mapping recruited 73.82 % of the reads for the newly sequenced sample, showing that these dereplicated genomes still explain most of the diversity in the geyser (compared to an average of 51.36 % for the samples from 2018). While all the genomes abundant in 2018 were still abundant in 2019, with *Altiarchaeales* is still being the only archaeon and by far the most abundant microorganism, and *Caldisericum* being the most abundant bacterium, five out of the 14 dereplicated genomes had no coverage in 2019, indicating that they either were transient members of the community or possibly were no longer sampled using the new sampling method. One organism that was still recovered but at a much lower abundance is the genome *Proteobacteria_59_48*, whose coverage decreased from around 50 to just three.

Thus to sum up, the abundant members of the community seem to be stable based on sequencings from two consecutive years, while low abundant community members seem to be more transient (though that might also be caused by the alternate sampling method).

Regarding the extent to which the sampled community represents the Andernach ecosystem:

Because the communities sampled at Geyser Andernach are planktonic, the extent to which they represent the entire in situ community (including biofilms) is not known. This point should be acknowledged.

Response 47: We have specified that only the planktonic fraction of microorganisms was sampled. Please see L110-112, stating:

“To investigate the community in subsurface fluids impacted by mantle degassing, we sampled two eruptions of Geyser Andernach, and collected the planktonic fraction of microorganisms onto three individual 0.1- μm filters.”

The “reconstructed genome” is said to be “representative for the ecosystem”. This statement needs to be more carefully phrased, because the taxonomic composition of the ecosystem may change over time (above) and the samples are limited to planktonic organisms. It can be finessed by saying that the reconstructed genome is representative of the planktonic ecosystem at the time of sampling.

Response 48: We have incorporated a new statement in L115-118:

“Approximately 75% of the reads (2.6% SD) mapped back the assembly providing evidence that the reconstructed metagenome is representative for the planktonic community at the time of sampling.”

Regarding in situ replication:

“In situ replication indices” are central to the manuscript. However, they’re not explained anywhere in the manuscript. It appears to me that the manuscript is using the “Index of Replication (iRep) metric introduced by Brown et al. (2016). As described by Brown et al., iRep “...determines replication rates based on measuring the rate of the decrease in average sequence coverage from the origin to the terminus of replication. ... the iRep algorithm is

distinct in that it makes use of the total change in coverage across all genome fragments”.

Based on my understanding of Brown et al. (2016), the discussion of replication in the present manuscript needs to be edited to account for the following points:

1. The technique must be clearly described before the data are introduced or interpreted. This description could be as brief as the lines from Brown that I quote above.

Response 49: We have added a description of how replication index values are calculated in the results (L130-133) and in the methods (L437-446), stating respectively:

“Replication index values are calculated from the difference of sequencing coverage between the origin of replication and terminus of replication. Proliferating organisms replicate their genomes with multiple replication forks starting at the replication origin and thus contributing more to sequencing reads.”

“The calculation of *in situ* replication index values is based on the assumption that organisms, that are actively proliferating, replicate their genome starting at the origin of replication and ending at the terminus of replication. Replicating organisms can thus have already replicated the parts of their genome close to the origin of replication but have not yet completed replicating sequences close to the terminus of replication. This can result in higher relative coverage of the sequence close to the origin of replication compared to the terminus of replication. Multiple simultaneous replication processes can exuberate this difference further. The *In situ* index of replication (iRep) estimates the number of replication processes based on this coverage difference but only works in Bacteria as Archaea can have multiple origins of replication³¹ and thus the iRep signal is distorted and cannot be applied in a comparative manner.”

We additionally added a supplementary discussion section about the iRep calculation and what can influence it as well as what is done in this work to mitigate these intervening effects (L186-218).

2. The result is a single index (not indices) and should be identified as such.

Response 50: When talking about a single iRep value, we used “index”. In the case of multiple iRep values, we have rephrased “indices” to “index values”.

3. Despite the lines from Brown quoted above, the index measures a ratio, not a rate. In short, it provides an estimate of the proportion of cells that is replicating DNA. It does not provide an estimate of how long that replication takes. In principal, it seems to me that this replication could take years (or even never reach completion). This point should be briefly discussed.

Response 51: We have rephrased that iRep does not represent an activity or rate (regarding with a time). Please see response 2 for a list of changes of the iRep component of the manuscript.

Specific examples of this issue can be found in the lines referenced below.

Lines 127-128 – “in situ replication indices” are introduced without definition or explanation and interpreted to demonstrate that 40-50% of the population was dividing. Because this is a fundamental claim of the manuscript, what’s really being measured should be made clear before the claim of reproduction is made.

In lines 190-219, “in situ replication indices” are referenced as iRep and plotted in figure 3 but still undefined, and this claim of subsurface replication is extended to sites in different regions of Earth.

This problem is amplified by the fact that iRep is not even defined in the Methods, but simply referenced there to another publication that does not address indices of replication (Guindon et al., 2010)! The correct reference should be 40 (Brown et al., 2016).

Response 52: We would like to point to response 49 for the respective results and methods sections introducing replication index values and the iRep abbreviation. We apologize for the citation miscall.

Lines 282-284 – Sedimentary cells are not necessarily “sessile” (attached) and comparison to a few shallow sediment samples (Figure 4) does not necessarily “validate that replication indices can be used as proxies of activity in these [active aquifer] ecosystems.”

Response 53: Since previous studies documented higher activity in sediments compared to aquifers, we changed our wording a voided stating that organisms are attached. However, our comparison of sediment and aquifer iRep values is thus still valid. Nevertheless, we agree that there is just one sampling site available for this analysis. We checked public databases and could not find any other sediment versus aquifer study that genomically resolved the community to further test our statement. We consequently toned down the iRep results as requested by other reviewers as well.

References

1. Bornemann, T. L. V., Esser, S. P., Stach, T. L., Burg, T. & Probst, A. J. uBin – a manual refining tool for metagenomic bins designed for educational purposes. *bioRxiv* 2020.07.15.204776 (2020) doi:10.1101/2020.07.15.204776.
2. Long, A. M., Hou, S., Ignacio-Espinoza, J. C. & Fuhrman, J. A. Benchmarking microbial growth rate predictions from metagenomes. *The ISME Journal* **15**, 183–195 (2021).
3. Mehrshad, M. *et al.* Energy efficiency and biological interactions define the core microbiome of deep oligotrophic groundwater. <http://biorxiv.org/lookup/doi/10.1101/2020.05.24.111179> (2020) doi:10.1101/2020.05.24.111179.
4. Kirkpatrick, J. B., Walsh, E. A. & D'Hondt, S. Microbial Selection and Survival in Subseafloor Sediment. *Front. Microbiol.* **10**, (2019).
5. Lloyd, K. G. *et al.* Evidence for a growth zone for deep subsurface microbial clades in near-surface anoxic sediments. *bioRxiv* 2020.03.24.005512 (2020) doi:10.1101/2020.03.24.005512.
6. Starnawski, P. *et al.* Microbial community assembly and evolution in subseafloor sediment. *Proc Natl Acad Sci USA* **114**, 2940–2945 (2017).
7. Lloyd, K. G. *et al.* Evidence for a Growth Zone for Deep-Subsurface Microbial Clades in Near-Surface Anoxic Sediments. *Appl. Environ. Microbiol.* **86**, (2020).
8. D'Hondt, S., Pockalny, R., Fulfer, V. M. & Spivack, A. J. Subseafloor life and its biogeochemical impacts. *Nature Communications* **10**, 3519 (2019).
9. Tully, B. J., Wheat, C. G., Glazer, B. T. & Huber, J. A. A dynamic microbial community with high functional redundancy inhabits the cold, oxic subseafloor aquifer. *The ISME Journal* **12**, 1 (2017).

10. Probst, A. J. *et al.* Biology of a widespread uncultivated archaeon that contributes to carbon fixation in the subsurface. *Nature Communications* **5**, 5497 (2014).
11. Becraft, E. D. *et al.* Evolutionary stasis of a deep subsurface microbial lineage. *The ISME Journal* 1–13 (2021) doi:10.1038/s41396-021-00965-3.
12. Cocks, L. R. M. & Torsvik, T. H. Baltica from the late Precambrian to mid-Palaeozoic times: The gain and loss of a terrane's identity. *Earth-Science Reviews* **72**, 39–66 (2005).
13. Torsvik, T. H. *et al.* Phanerozoic polar wander, palaeogeography and dynamics. *Earth-Science Reviews* **114**, 325–368 (2012).
14. Maruyama, S., Isozaki, Y., Kimura, G. & Terabayashi, M. Paleogeographic maps of the Japanese Islands: Plate tectonic synthesis from 750 Ma to the present. *Island Arc* **6**, 121–142 (1997).
15. Hausner, M. & Wuertz, S. High rates of conjugation in bacterial biofilms as determined by quantitative in situ analysis. *Appl Environ Microbiol* **65**, 3710–3713 (1999).
16. Nelson-Sathi, S. *et al.* Origins of major archaeal clades correspond to gene acquisitions from bacteria. *Nature* **517**, 77–80 (2015).
17. Aiuppa, A., Fischer, T. P., Plank, T. & Bani, P. CO₂ flux emissions from the Earth's most actively degassing volcanoes, 2005–2015. *Scientific Reports* **9**, 5442 (2019).
18. Bräuer, K., Kämpf, H., Niedermann, S. & Strauch, G. Indications for the existence of different magmatic reservoirs beneath the Eifel area (Germany): A multi-isotope (C, N, He, Ne, Ar) approach. *Chemical Geology* **356**, 193–208 (2013).
19. Werner, C. *et al.* Carbon Dioxide Emissions from Subaerial Volcanic Regions: Two Decades in Review. in *Deep Carbon* (eds. Orcutt, B. N., Daniel, I. & Dasgupta, R.) 188–236 (Cambridge University Press, 2019). doi:10.1017/9781108677950.008.

20. Caracausi, A. & Paternoster, M. Radiogenic helium degassing and rock fracturing: A case study of the southern Apennines active tectonic region. *Journal of Geophysical Research: Solid Earth* **120**, 2200–2211 (2015).
21. Loreto, M. F., Italiano, F., Deponte, D., Facchin, L. & Zgur, F. Mantle degassing on a near shore volcano, SE Tyrrhenian Sea. *Terra Nova* **27**, 195–205 (2015).
22. Gilfillan, S. M. V. *et al.* Noble gases confirm plume-related mantle degassing beneath Southern Africa. *Nature Communications* **10**, 1–7 (2019).
23. Lee, H. *et al.* Mantle degassing along strike-slip faults in the Southeastern Korean Peninsula. *Scientific Reports* **9**, 1–9 (2019).
24. Fullerton, K. M. *et al.* Plate tectonics drive deep biosphere microbial community composition. <https://osf.io/gyr7n> (2019) doi:10.31223/osf.io/gyr7n.
25. Chaumeil, P.-A., Mussig, A. J., Hugenholtz, P. & Parks, D. H. GTDB-Tk: a toolkit to classify genomes with the Genome Taxonomy Database. *Bioinformatics* **36**, 1925–1927 (2020).
26. Anantharaman, K. *et al.* Thousands of microbial genomes shed light on interconnected biogeochemical processes in an aquifer system. *Nature Communications* **7**, 1–11 (2016).
27. Kallmeyer, J., Pockalny, R., Adhikari, R. R., Smith, D. C. & D’Hondt, S. Global distribution of microbial abundance and biomass in subseafloor sediment. *Proc Natl Acad Sci U S A* **109**, 16213–16216 (2012).
28. Wood, H. G. Life with CO or CO₂ and H₂ as a source of carbon and energy. *The FASEB Journal* **5**, 156–163 (1991).
29. Beulig, F. *et al.* Carbon flow from volcanic CO₂ into soil microbial communities of a wetland mofette. *ISME J* **9**, 746–759 (2015).
30. Beulig, F. *et al.* Altered carbon turnover processes and microbiomes in soils under long-term extremely high CO₂ exposure. *Nature Microbiology* **1**, 1–10 (2016).

31. Wu, Z., Liu, J., Yang, H. & Xiang, H. DNA replication origins in archaea. *Front Microbiol* **5**, (2014).

REVIEWERS' COMMENTS

Reviewer #2 (Remarks to the Author):